# Root Cause Analysis of Outliers with Missing Structural Knowledge

**William Roy Orchard***
University of Cambridge
Cambridge, UK
wo223@cam.ac.uk

**Nastaran Okati***
Max Planck Institute for Software Systems
Kaiserslautern, Germany

**Sergio Hernan Garrido Mejia**
Max Planck Institute for Intelligent Systems
Amazon
Tübingen, Germany

**Patrick Blöbaum**
Amazon
Tübingen, Germany

**Dominik Janzing**
Amazon
Tübingen, Germany

## Abstract

The goal of Root Cause Analysis (RCA) is to explain why an anomaly occurred by identifying where the fault originated. Several recent works model the anomalous event as resulting from a change in the causal mechanism at the root cause, i.e., as a soft intervention. RCA is then the task of identifying which causal mechanism changed. In real-world applications, one often has either few or only a single sample from the post-intervention distribution: a severe limitation for most methods, which assume one knows or can estimate the distribution. However, even those that do not are statistically ill-posed due to the need to probe regression models in regions of low probability density. In this paper, we propose simple, efficient methods to overcome both difficulties in the case where there is a single root cause and the causal graph is a polytree. When one knows the causal graph, we give guarantees for a traversal algorithm that requires only marginal anomaly scores and does not depend on specifying an arbitrary anomaly score cut-off. When one does not know the causal graph, we show that the heuristic of identifying root causes as the variables with the highest marginal anomaly scores is causally justified. To this end, we prove that anomalies with small scores are unlikely to cause those with larger scores in polytrees and give upper bounds for the likelihood of causal pathways with non-monotonic anomaly scores.

## 1 Introduction

When an anomaly occurs, it is essential to identify its root cause so that action can be taken to resolve it. As such, Root Cause Analysis (RCA), has become a rich and rapidly developing area of research, with wide ranging applications from meteorology [1], monitoring health [2, 3], industrial fabrication [4], fraud detection [5], credit scoring [6], cloud applications [7, 8, 9, 10, 11, 12] and more [13, 14, 15, 16, 17]. Although many previous works have adopted a purely statistical approach to RCA [18, 19, 20, 21, 22], relying on correlations, the fact that the goal is to take *action* distinguishes

---

*Authors contributed equally.

39th Conference on Neural Information Processing Systems (NeurIPS 2025).

it as a necessarily causal problem. The predominant choice in causal literature is to model the observed anomaly as being the result of a change in the causal mechanism at the root cause, i.e., as a soft intervention. RCA can then be broadly understood as the task of identifying the variable whose mechanism changed. Budhathoki et al. [23] give a formalisation of RCA as a quantitative contribution analysis based on counterfactuals – asking, "would the observed anomaly still be anomalous had this mechanism been as normal?" – but its dependence on knowing the full structural causal model is a severe bottleneck to its practical application. As such, from a causal perspective, work on RCA is broadly categorised into two groups: those that require the causal DAG, and those that do not or try to infer it from data. With the causal DAG, a widely used class of methods is traversal-based, as in [24, 25, 9, 26], wherein a variable is identified as a root cause if it is anomalous, none of its parents are anomalous, and it is linked to the target variable through a path of anomalous variables. For example, Hardt et al. [27] apply such a traversal algorithm to data from cloud computing to identify the root cause of performance drops in a microservice-based application [12, 28, 29]. When the causal DAG is unknown, the main idea behind the second line of work is that the anomalies themselves can support causal discovery. For example, Ikram et al. [10] treat RCA as a causal discovery problem where the goal is to find the target(s) of a soft intervention [30]. Likewise, [31, 32] describe how to use heavy-tailed distributions to infer the causal DAG for linear models. In a sense, this also amounts to using anomalies for causal discovery – if one counts points in the tail as anomalies.

## 1.1 Our contributions

Most methods assume one knows or can estimate the post-intervention, "anomalous", distribution, but in real-world scenarios, it is not uncommon to have only a single sample from the distribution. In business applications, faults must be resolved rapidly, leaving no time to collect further samples. In personalised medicine, the goal is to treat each individual patient according to the unique root causes of their disease, and so while one has many samples of the healthy population, one assumes a priori that each diseased patient is unique. Even those methods that do not assume access to more than one sample from the anomalous distribution are often statistically ill-posed due to the need to estimate causal conditional probabilities or else to probe regression models in regions with low probability density. In this paper, we argue that these challenges can be overcome when the task is to identify a single root cause, and propose two simple and efficient methods for doing so. In section 2, we give the preliminaries to causal RCA and introduce 'information theoretic' (IT) anomaly scores upon which our methods are based. In section 3 we show that marginal anomaly scores are already sufficient for drawing conclusions about the causal relationships between anomalies: first for cause-effect pairs and then more generally when the causal DAG is a polytree. We show that marginal anomaly score 'jumps' along causal paths can enable identification of the root cause and propose SMOOTH TRAVERSAL, for when one knows the causal graph. We conclude Section 3 by arguing that the heuristic of selecting the variables with the highest IT anomaly scores as candidate root causes is causally justified in polytrees, and propose SCORE ORDERING, for when one does not know the causal graph. We also discuss its theoretical limitations. Finally, in Section 4 we compare existing approaches to RCA to our own proposals on both synthetic and real-world data, demonstrating that despite their simplicity, our approaches are competitive. To our knowledge, no other methods operate with causal guarantees in an RCA setting as general as we address in this paper.

## 2 Preliminaries

We consider the scenario in which the value $x_n$ of a target variable $X_n$ has been flagged as an anomaly by an anomaly detection algorithm. We have jointly observed values $(x_1, \ldots, x_n)$ of variables $(X_1, \ldots, X_n)$. Our goal is to identify a unique root cause of the anomaly $x_n$ from among the variables $X_1, \ldots, X_n$.

We assume the causal relationships between $X_1, \ldots, X_n$ are given by a Causal Bayesian Network (CBN) [33], so that their joint distribution factorises into causal conditionals according to an underlying causal DAG:

$$P(X_1, \ldots, X_n) = \prod_{i=1}^{n} P(X_i \mid PA_i), \tag{1}$$

where each $P(X_i \mid PA_i)$ corresponds to the causal mechanism of variable $X_i$ given its parents $PA_i$ in the causal DAG. We then model the anomaly $x_n$ as having arisen due to one of the causal

mechanisms having been 'corrupted'. We observe only the single sample $(x_1, \ldots, x_n)$ drawn from the 'anomalous' distribution

$$\tilde{P}(X_1, \ldots, X_n) = \tilde{P}(X_j \mid PA_j) \prod_{i \neq j}^{n} P(X_i \mid PA_i), \tag{2}$$

wherein the 'normal' mechanism $P(X_j \mid PA_j)$ has been replaced by a corrupted one $\tilde{P}(X_j \mid PA_j)$ at the root cause. Identifying the root cause is then a matter of identifying which causal conditional changed from the normal period based on a single sample from the anomalous one. While we can never exclude the case where several mechanisms are corrupted, we always start with the working hypothesis that there has been only one. If this does not work, we can still try hypotheses with more than one, but with the strong inductive bias of preferring explanations where most mechanisms worked as expected, in agreement with the so-called 'sparse mechanism shift hypothesis' [34, 35]. It is also important to note that although we have adopted the language of CBNs, our results can be recast in terms of the counterfactual contribution analysis introduced in [23] (see Appendix A). Indeed, it is noteworthy in itself that the counterfactual description can be simplified to an interventional one for the case of identifying a single root cause.

## 2.1 Anomaly score

We next define an anomaly score, which we use in the following sections. Let $\tau : \mathcal{X} \to \mathbb{R}$ be an appropriate feature map, which can be any existing anomaly score function, mapping elements of $\mathcal{X}$ to real values, for example, the z-score. Further, define the event

$$E := \{\tau(X_n) \geq \tau(x_n)\}, \tag{3}$$

of being at least as extreme as the observed event $x_n$ according to the feature function $\tau$. As the goal of RCA is to identify which causal mechanism has been corrupted, an important interpretation of $\tau$ is as a test statistic for the null hypothesis that $x_n$ was drawn from $P(X_n)$, i.e., $H_0 : X_n \sim P(X_n)$, with $P(\tau(X_n) \geq \tau(x_n)) = P(E)$ as the corresponding p-value. Small $P(E)$ indicates that the null hypothesis is rejected with high confidence[2] and we assume rather that the usual mechanism generating samples from $P(X_n)$ has been corrupted for that specific statistical unit. Since small p-values correspond to strong anomalies, we define the *marginal* anomaly score by

$$S(x_n) := -\log P(\tau(X_n) \geq \tau(x_n)) = -\log P(E). \tag{4}$$

Since the logarithm of the probability of an event measures its *surprise* in information theory [13, 15], as in [23], we call anomaly scores with the above calibration *IT scores* for *information theoretic*. Given $k$ observations $x_n^1, \ldots, x_n^k$, including the single anomalous one, we will use the following simple estimator[3]

$$\hat{S}(x_n^j) := -\log \frac{1}{k} \left| \{i \in \{1, \ldots, k\} : \text{with } \tau(x_n^i) \geq \tau(x_n^j)\} \right|. \tag{5}$$

It is important to note that this value can be at most $\log k$, i.e., the estimated score can attain the value $s$ only when at least $e^s$ samples have been used. We provide detailed sample complexity results for IT anomaly score estimation in Appendix B.

In addition to the marginal anomaly score, we also introduce the *conditional* IT anomaly score (as in the `arXiv` version of [23]):

$$S(x_n \mid pa_n) := -\log P(\tau(X_i) \geq \tau(x_n) \mid PA_n = pa_n), \tag{6}$$

where now $\tau$ becomes a test statistic for the null hypothesis that $x_n$ was drawn according to its causal mechanism $P(X_n \mid PA_n = pa_n)$ for that particular statistical unit. Although suppressed notationally, we allow variable-dependent feature functions $\tau_i$, but they must coincide between marginal and conditional anomaly scores. As the goal of RCA, as stated, is to identify which causal mechanism was corrupted, the conditional anomaly score is instrumental to what follows. To illustrate the difference between marginal and conditional anomaly scores, consider the following example:

---

[2]This interpretation is certainly invalid in a standard anomaly *detection* scenario where Eq. 3 is repeatedly computed for every observation in a sequence. However, so-called anytime p-values [36] are outside of the scope of this paper.

[3]This estimator is not just a finite sample approximation of (4), but also the negative logarithm of a conformal p-value [37] for the null hypothesis of exchangeability.

**Example 1** (Linear cause-effect model). *Consider causal model $Y := \beta X + N$, $N \perp\!\!\!\perp X$. For strictly increasing $\tau$ (e.g. the z-score, $\tau(y) = (y - \mu_Y)/\sigma_Y$), and observation $(x, y)$, the marginal anomaly score $S(y) = -\log P(Y \geq y)$ is simply the 'surprise' of the one-sided tail event for $Y$. However, the conditional anomaly score $S(y \mid x) = -\log P(Y \geq y \mid X = x) = -\log P(bx + N \geq y) = -\log P(N \geq y - bx)$, is a marginal anomaly score of the* residual *after accounting for $X$.*

For the following sections, all proofs can be found in Appendix C unless otherwise indicated.

## 3   Root cause analysis when conditional probability estimation is ill-posed

To answer the question of which causal mechanism was corrupted for the sample $(x_1, \ldots, x_n)$, we need to determine which value $x_i$ cannot be 'explained' by the values of its parents according to its normal causal conditional, $P(X_i \mid PA_i)$. However, inferring the causal conditional is statistically ill-posed, particularly when the number of parents is large. One could, for example, make the additive noise assumption [38, 39] and thereby reduce the problem to that of estimating conditional expectations, but the problem remains ill-posed because analysing anomalies amounts to probing regression models in regions with low probability density. In this section, we show that causal relationships can be inferred between anomalies with a given causal DAG together with only *marginal* anomaly scores $\{S(x_i), \ldots, S(x_n)\}$, using the simple estimator in Eq. 5, in the case the causal DAG is a polytree. However, to introduce the ideas we will need, we start first with cause-effect pairs, then discuss the problem more generally in DAGs, and then lastly restrict our attention to polytrees.

### 3.1   Anomalies in cause and effect pairs

Let the causal DAG be $X \to Y$ for two variables $X, Y$, and we observe $(x, y)$ with anomaly scores $S(x), S(y)$. We define two null hypotheses that state that each causal mechanism, $P(X)$ or $P(Y \mid X)$, worked as normal: $H_0^X$: $x$ was drawn from $P(X)$, and $H_0^Y$: $y$ was drawn from $P(Y \mid X = x)$. Note that $H_0^Y$ does not impose a constraint on $X$, it allows $x$ to be drawn from an arbitrary distribution instead of $P(X)$, only the mechanism generating $y$ from $x$ is assumed to have remained 'normal'. We show that whether one should reject one or both hypotheses can be determined from a comparison of $S(x)$ and $S(y)$ alone. To see how, we first need to introduce a definition which we will justify shortly:

**Definition 3.1** (Bivariate score typicality). *We say that observation $(x, y)$ for $X, Y$ satisfies score typicality if*

$$S(y \mid x) \geq |S(y) - S(x)|_+, \tag{7}$$

*where $|\cdot|_+$ denotes the positive part.*

We then have the following criteria for rejecting $H_0^X$ and $H_0^Y$:

**Lemma 3.2** (p-value bound for marginal anomaly event.). *$H_0^X$ can be rejected at level $p \leq e^{-S(x)}$.*

This follows immediately from the definition of IT scores in Eq. 4. On the other hand, we obtain:

**Lemma 3.3** (Anomalies rarely cause larger anomalies). *Subject to score typicality, $H_0^Y$ can be rejected at level $p \leq e^{-|S(y)-S(x)|_+}$.*

*Proof.* That $H_0^Y$ can be rejected at level $p = e^{-S(y|x)}$ follows from the definition of conditional IT scores in Eq. 6, the result then follows immediately, subject to score typicality. $\square$

Lemma 3.3 therefore states that, subject to score typicality, an anomaly of strength $S(x)$ causes an anomaly of strength $S(y)$ only with probability at most $e^{-|S(y)-S(x)|_+}$, i.e. an anomaly at $X$ is unlikely to cause a much stronger anomaly at $Y$ (see also Corollary 3.3 in [40] for an algorithmic information theoretic view of this result).

Score typicality, therefore, gives us a criterion for rejecting $H_0^Y$ using only marginal scores, but the question remains whether we should expect it to be satisfied. The following lemma shows that for any given $y$, although we cannot guarantee score typicality is satisfied for any *particular* $x$, we can conclude that the bound is likely to hold approximately for a randomly chosen $x$:

**Lemma 3.4** (Score typicality probably holds approximately). *Let the distribution of $X$ (possibly multivariate) be absolutely continuous with respect to the Lebesgue measure. For any $s_X \leq S(y) - 1$,*

*let x be chosen at random according to $P\big(X \mid S(X) \in [s_X, S(y)]\big)$. Then we obtain, with probability at least $1 - 2/c$, an x for which the following inequality holds:*

$$S(y \mid x) \geq |S(y) - S(x)|_+ - 2\log c. \tag{8}$$

There are also non-trivial cases where score typicality holds exactly:

**Lemma 3.5** (Injectivity and monotonicity imply score typicality). *Let the mapping $x \mapsto S(x)$ be one-to-one, and $P(S(Y) \geq S(y) \mid S(X) = S(x)) \leq P(S(Y) \geq S(y) \mid S(X) \geq S(x))$. Then observation $(x, y)$ satisfies score typicality.*

The first condition is the injectivity of $S$. The second is a monotonicity condition, it says: increasing the anomaly score of $X$ does not decrease the probability of an anomaly event at $Y$. Both assumptions are satisfied for all $(x, y)$, for example, for real-valued variables when $Y := f(X, N)$, where $N$ is an independent noise term, $f$ is monotonic in $X$ and $N$ and $\tau = id$.

In the following sections, we state our results subject to score typicality for ease of exposition, but we should keep in mind that Lemmas 3.4 and 3.5 show that we could avoid it entirely, at the expense of having slightly weaker bounds holding for most samples, or by assuming injectivity and monotonicity.

Returning to our bivariate case, there is also a third hypothesis, namely that $x$ and $y$ are causally unrelated. Even if there is a causal connection between $X$ and $Y$, in this case $X \to Y$, it could happen that $x$ being anomalous was irrelevant for the anomaly $y$ because $P(Y|X = x) = P(Y)$. In other words, although $x$ is anomalous, it happens to be an anomaly that does not render an anomaly at $Y$ more likely. The corresponding null hypothesis is $H_0^{XY}$: $x$ was drawn from $P(X)$ and $y$ from $P(Y)$. We find:

**Lemma 3.6** (Bound on $p$-value for independent anomalies). *$H_0^{XY}$ can be rejected at level $p \leq e^{-S(x)-S(y)}[1 + S(x) + S(y)]$.*

The lemma follows from the fact that for independent variables $X, Y$, the corrected sum $S(x)+S(y)-\log[1 + S(x) + S(y)] =: \tilde{S}((x, y))$ is a valid IT score for pairs and thus satisfies $P(\tilde{S}((X, Y)) \geq \tilde{S}((x, y))) \leq e^{-\tilde{S}((x,y))}$ (see Theorem 1 in the `arXiv` version of [23]).

Lemmas 3.2 to 3.6 illustrate some of the ideas that will be of use in the case the causal graph is a polytree: in particular, Lemma 3.3 is noteworthy as it shows that we can test hypotheses concerning conditional distributions using differences between *marginal* anomaly scores, so long as we have the causal graph. However, they also entail interesting consequences for the case where the causal direction is not known. For example, let $S(x) = 10, S(y) = 5$ and significance level $\alpha = 0.01$. For $X \to Y$ we can only reject $H_0^X$, while the mechanism $P(Y|X)$ possibly worked as expected. However, for $Y \to X$, we would reject both hypotheses, that $P(Y)$ and that $P(X|Y)$ worked as expected with $p$-value $e^{-5}$ each. Following the working hypothesis that at most one mechanism was corrupted, we thus prefer $X \to Y$. Note that the third alternative of causally independent generation of $x$ and $y$ can be rejected even at level $p \leq e^{-5-10}(1 + 5 + 10)$.

## 3.2 Anomalies in DAGs

Before restricting our attention to the case where the causal DAG is a polytree, let us first consider the problem in full generality in DAGs. To infer for which variable the causal mechanism has been corrupted, we start with the hypothesis that all mechanisms worked as expected except for $P(X_j \mid PA_j)$: $H_0^j$: for the anomaly event $(x_1, \ldots, x_n)$ all $x_i$ with $i \neq j$ have been drawn from $P(X_i \mid PA_i = pa_i)$. We will return to marginal anomaly scores shortly, but first, to define test statistics, we focus on conditional anomaly scores. We also need to make the assumption:

1. **Continuity:** all variables $X_i$ are continuous with density w.r.t. the Lebesgue measure, and also all conditional distributions $P(X_i|PA_i = pa_i)$ have such a density.

Referring to Eq. 6, conditional scores with variable inputs $X_i, PA_i$ define random variables $S(X_i \mid PA_i)$. Continuity ensures that all conditional anomaly scores are distributed according to the density $p(s) = e^{-s}$ for $s \geq 0$, and entails a property that will be convenient for testing $H_0^j$:

**Lemma 3.7** (Conditional anomaly scores are independent). *Subject to the continuity assumption, $\{S(X_1|PA_1), \cdots, S(X_n|PA_n)\}$ are independent random variables.*

Independence of conditional scores enables the definition of a simple test statistic that is obtained by summing over them[4] together with a correction term. With $S_{\text{sum}} := \sum_{i \neq j} S(X_i | PA_i)$, we define the joint anomaly score

$$S := S_{\text{sum}} - \log \sum_{l=0}^{n-2} \frac{S_{\text{sum}}^l}{l!}. \tag{9}$$

To understand Eq. 9, note that $S_{\text{sum}}$ no longer has the properties of an IT anomaly score: the sum of independent IT scores is likely to be large because it is not unlikely that the set contains at least one large term. The second term is therefore needed to 'recalibrate' the sum and is akin to a multiple testing correction. The following result states that this is quantitatively the right correction:

**Lemma 3.8** (The joint score is an IT anomaly score). *If $H_0^j$ holds, Eq. 9 is distributed according to the density $p(s) = e^{-s}$ for $s \geq 0$.*

As a direct result of the above lemmas, we have:

**Theorem 3.9** (*$p$-value for joint anomaly event*). *$H_0^j$ can be rejected for the observation $(x_1, \ldots, x_n)$ with $p$-value $p = e^{-S_{\text{sum}}} \cdot \sum_{l=0}^{n-2} \frac{S_{\text{sum}}^l}{l!}$.*

The theorem justifies choosing the index $j$ with the maximal conditional anomaly score as the root cause, whenever we follow the working hypothesis that only one mechanism is corrupted.

With our framework in place, we can return to our original goal: to show how RCA is possible, without the estimation of conditional probabilities, using only marginal anomaly scores. In order to do so, we restrict our attention to the case where the causal DAG is a polytree.

### 3.3 Anomalies in polytrees

Polytrees are DAGs where the underlying skeleton is a tree (see Fig. 2 in Appendix D). The key property of polytrees that we exploit is that the parents of any given variable are marginally independent. This allows us to replace conditional anomaly scores with bounds derived from *marginal* anomaly scores, following the ideas from the bivariate case. To this end, note that, as they are independent, we may use the same construction as we did in Eq. 9 to define a joint score over the parents of a given variable. For variable $X_i$, with parents $PA_i^1, \ldots, PA_i^k$, and event $pa_i$:

$$S_{\text{joint}}(pa_i) := \sum_{j=1}^{k} S(pa_i^j) - \log \sum_{l=0}^{k-1} \frac{(\sum_{j=1}^{k} S(pa_i^j))^l}{l!}. \tag{10}$$

That this is indeed an IT score for joint event $pa_i$ follows from the same proof as for Lemma 3.8. For brevity, we drop the subscript and simply write $S(pa_i)$ to mean the joint anomaly score of multivariate event $pa_i$, as it will be clear from context whether we refer to a marginal score or a joint score. By using the joint score over the parents of each variable, we have effectively reduced the polytree case to the bivariate one with respect to each variable and its parents. In particular, we generalise score typicality:

**Definition 3.10** (Score typicality). *We say that observation $(x, pa_i)$ for $X, PA_i$ satisfies score typicality if*

$$S(x_i \mid pa_i) \geq |S(x_i) - S(pa_i)|_+. \tag{11}$$

Subject to the continuity assumption, Lemma 3.4 applies directly, in statement and proof, to score typicality as defined above, simply substituting $Y$ for $X_i$ and $X$ for $PA_i$. We can therefore say that for observation $x_i$, score typicality is likely to hold approximately for a randomly chosen $pa_i$, in the sense given by Lemma 3.4.

As a result, we obtain the following bound on the joint statistics introduced in the previous subsection:

**Theorem 3.11.** (Bound on $p$-value for joint anomaly event using only marginal anomaly scores). *With $\hat{S}_{\text{sum}} = \sum_{i \neq j} |S(x_i) - S(pa_i)|_+$ it holds that subject to score typicality, $H_0^j$ can be rejected with $p$-value*

$$p \leq e^{-\hat{S}_{\text{sum}}} \cdot \sum_{l=0}^{n-2} \frac{\hat{S}_{\text{sum}}^l}{l!}. \tag{12}$$

---

[4]Note that this corresponds to Fisher's method [41] for aggregating $p$-values, which also considers the sum of independent, log-transformed $p$-values.

We conclude that $H_0^j$ needs to be rejected whenever the anomaly score increases significantly along a path of anomalies at any $i \neq j$. This justifies inferring the index $j$ that maximises the score difference $|S(x_j) - S(pa_j)|_+$ as the unique root cause (with the difference being $|S(x_i) - 0|_+ = S(x_i)$ for any node with no parents) because this yields the weakest bound in Eq. 12. Motivated by these bounds, we propose the algorithm SMOOTH TRAVERSAL[5] which selects the node as the root cause that shows the strongest increase in anomaly score compared to its parents (as a heuristic we consider the score difference with the most anomalous parent of each node, see Algorithm 1 below). In the usual traversal approach [24, 25, 9], a variable is identified as a root cause if it is anomalous, none of its parents are, and it is linked to the target variable through a path of anomalous variables. However, this requires the user to choose a threshold above which a node is considered anomalous. SMOOTH TRAVERSAL avoids having to make any such arbitrary choice. We can prove that if the maximum score difference is large, then the node at which the difference is maximised is likely to be the root cause. To this end: for anomaly event $(x_1, \ldots, x_n)$ and anomaly score differences

$$\delta_i := \big|S(x_i) - S(pa_i)\big|_+, \quad i = 1, \ldots, n,$$

define null hypothesis $H_0^{\max}$: for anomaly event $(x_1, \ldots, x_n)$ and $j = \mathrm{argmax}_i\, \delta_i$, $x_j$ was drawn from $P(X_j \mid PA_j = pa_j)$, i.e. the anomaly with the largest score difference with its parents is not the root cause.

**Theorem 3.12** (Bound on $p$-value for anomaly with biggest jump not being the root cause). *Subject to score typicality, when there is a single root cause, $H_0^{\max}$ can be rejected for observation $(x_1, \ldots, x_n)$, and maximum score difference $\delta_{\max} := \max(\delta_1, \ldots, \delta_n)$ with p-value*

$$p \leq 1 - (1 - e^{-\delta_{\max}})^{n-1}. \tag{13}$$

The theorem follows from the fact that if the node maximising the score difference from its parent is not the root cause, it implies that at least one conditional anomaly score, of the $(n-1)$ non-root causes, is at least $\delta_{\max}$.[6]

---

**Algorithm 1 (Smooth Traversal)** Returns the variable with the largest positive score difference to its highest scoring parent

---

**Require:** Scores $S(x_1), \ldots, S(x_n)$, causal DAG $G$ of variables $X_1, \ldots, X_n$
**Ensure:** Variable $X_k$ with the largest positive score difference to its highest-scoring parent
 1: Initialize max_min_difference $\leftarrow -\infty$
 2: Initialize candidate_root_cause $\leftarrow$ None
 3: **for** each ancestor $X_i$ of $X_n$ in $G$ **do**
 4:     Let $\mathrm{PA}_i$ be the set of parents of $X_i$ in $G$
 5:     **if** $\mathrm{PA}_i \neq \emptyset$ **then**
 6:         max_parent_score $\leftarrow \max\limits_{X_j \in \mathrm{PA}_i} S(x_j)$
 7:     **else**
 8:         max_parent_score $\leftarrow 0$
 9:     **end if**
10:     score_difference $\leftarrow |S(x_i) - \text{max\_parent\_score}|_+$
11:     **if** score_difference $>$ max_min_difference **then**
12:         max_min_difference $\leftarrow$ score_difference
13:         candidate_root_cause $\leftarrow X_i$
14:     **end if**
15: **end for**
16: **return** candidate_root_cause

---

### 3.4 Shortlist of root causes via ordering anomaly scores

We have shown that a unique root cause can be identified using only marginal anomaly scores if the causal graph is known. We now drop the assumption that the causal graph is known and ask, can

---

[5]We thank Elke Kirschbaum for suggesting this name.

[6]The approach for deriving this $p$-value broadly resembles Tippett's method [42] for combining independent $p$-values.

we find the root cause given *only* the marginal anomaly scores $S(x_1), \ldots, S(x_n)$? We argue that the top-scoring anomalies constitute a good shortlist for the root cause.

To see this, assume again that the (unknown) causal DAG for variables $X_1, \ldots, X_n$ is a polytree as before, and for illustration, first consider a particular causal path through the graph,

$$X_{i_1} \to X_{i_2} \to \cdots \to X_{i_k}.$$

If we now 'coarsen' the graph, marginalising out all the intermediate variables between $X_{i_1}$ and $X_{i_k}$, so that we collapse the whole path into a single edge $X_{i_1} \to X_{i_k}$, application of Lemma 3.3 shows that we can reject the hypothesis that the new 'coarse' mechanism $P(X_{i_k} \mid X_{i_1} = x_{i_k})$ worked as expected at level $p \leq e^{-|S(x_{i_k}) - S(x_{i_1})|_+}$. However, note that the coarse mechanism for $X_{i_k}$ now implicitly encompasses each of the intermediate mechanisms that connected $X_{i_1}$ to $X_{i_k}$ in the full graph. In other words, the anomaly $x_{i_1}$ is unlikely to cause a much larger anomaly $x_{i_k}$ anywhere downstream, unless we allow that any of the intermediate mechanisms connecting them has been corrupted.

With this in mind, consider the full graph on $X_1, \ldots, X_n$ and let $\pi$ be a permutation of $\{1, \ldots, n\}$ such that

$$S(x_{\pi(1)}) \geq S(x_{\pi(2)}) \geq \cdots \geq S(x_{\pi(n)}).$$

We define the null hypothesis that the mechanisms worked as expected for all of the top-$k$ scoring anomalies: $H_0^{\text{top-}k}$: for anomaly event $(x_1, \ldots, x_n)$ all $x_{\pi(i)}$ with $i \leq k$ have been drawn from $P(X_{\pi(i)} \mid PA_{\pi(i)} = pa_{\pi(i)})$. Under the null hypothesis, we know that the score for $x_{\pi(1)}$ must have been greater by at least $\Delta_k := S(x_{\pi(1)}) - S(x_{\pi(k+1)})$ compared to an ancestral node, despite its mechanism, and the mechanisms of any intermediates, working as expected. This is because whether or not any of the variables $X_{\pi(2)}, \ldots, X_{\pi(k)}$ are intermediates, their mechanisms, by assumption, worked as expected under the null. If any of the variables $X_{\pi(k+2)}, \ldots, X_{\pi(n)}$ are intermediates (or if $X_{\pi(1)}$ has no ancestors), then its score increased by an even greater amount than $\Delta_k$. Applying the same coarsening idea as above, and accounting for multiple testing, we therefore have:

**Theorem 3.13.** (Bound on likelihood root cause not in top-$k$ highest scoring anomalies). *Subject to score typicality, where $d_{\max}$ denotes the maximum in-degree among all nodes in the graph, we can reject $H_0^{\text{top-}k}$ at level $p \leq n \cdot d_{\max} \cdot e^{-\Delta_k}$.*

These bounds motivate SCORE ORDERING which, given the marginal anomaly scores and an assumed upper bound on the in-degree of nodes in the causal graph, returns the set of top-$k$ highest scoring anomalies with the guarantee that the root cause is among them with at least the desired confidence level $1 - \alpha$ (see Algorithm 2). Note that one could just as well specify a desired number of variables, $k$, and SCORE ORDERING would then return a confidence level for each possible $d_{\max}$ so that the latter is not a strict input requirement.

---

**Algorithm 2 (Score Ordering)** Returns the top-$k$ scoring anomalies such that the root cause is among them with confidence at least $1 - \alpha$

---

**Require:** Scores $S(x_1), \ldots, S(x_n)$, maximum in-degree $d_{\max}$, confidence level $1 - \alpha$
**Ensure:** A set $L \subseteq \{X_1, \ldots, X_n\}$
1: Let $\pi$ be a permutation of $\{1, \ldots, n\}$ such that $S(x_{\pi(1)}) \geq S(x_{\pi(2)}) \geq \cdots \geq S(x_{\pi(n)})$
2: Initialize $L \leftarrow \emptyset$
3: **for** $k = 1$ to $n$ **do**
4:     $L \leftarrow L \cup \{X_{\pi(k)}\}$
5:     **if** $k = n$ **then**
6:         **return** $L$
7:     **end if**
8:     $\Delta \leftarrow S(x_{\pi(1)}) - S(x_{\pi(k+1)})$
9:     **if** $n \cdot d_{\max} \cdot e^{-\Delta} \leq \alpha$ **then**
10:        **return** $L$
11:     **end if**
12: **end for**

---

## 3.5 Beyond polytrees: limitations of score ordering

Most of the theory developed in this section has depended on the assumption that the causal graph is a polytree. While this assumption appears in causal literature (e.g. [43, 44, 45]), it is nonetheless a strong structural assumption. One of the strengths of both of the algorithms we propose, SMOOTH TRAVERSAL and SCORE ORDERING, is their simplicity and efficiency. SCORE ORDERING in particular demands very weak input requirements, and so the fact that it comes with any causal guarantees at all might be a surprise. As a minimum, this might justify its use as a 'first pass' heuristic in settings beyond those we study here. However, the question remains as to what extent we might expect our proposed algorithms to perform well in settings that do not meet the structural assumptions (we empirically assess this question in the next section). The key observation on which SCORE ORDERING depends in the proof of Theorem 3.13 is that an anomaly $x_i$ is unlikely to cause a much larger anomaly downstream. Li et al. [46] describe a condition under which the $z^2$-score of an effect $X_j$ can be larger than the $z^2$-score of the root cause $X_i$ in a DAG with linear structural equations. Although our results refer to IT scores rather than $z^2$-scores, their example nonetheless applies to us as the $z^2$-score can be turned into an IT score by a monotonic recalibration. They show (see Corollary 2.1 in [46]) that a sufficient condition for such a case *not* to occur is that the DAG be a polytree, and so their results do not contradict our own. However, in addition, we show in Appendix E that in DAGs (with linear structural equations) more generally such cases are 'rare' in the sense that, under reasonable assumptions, it will hold for only a 'small' fraction of root causes. Despite, then, exceptions occurring, we believe our results nonetheless give confidence that SCORE ORDERING can be applied as a useful first heuristic even beyond polytrees.

## 4 Experiments

We evaluate SMOOTH TRAVERSAL and SCORE ORDERING against a range of existing RCA methods, that can also operate with a single anomalous sample, on both synthetic and real world data. In particular we evaluate our methods against:

**'Cholesky'** [46] Assumes linear structural causal models, but does not require the causal graph. Makes use of the Cholesky decomposition of the covariance matrix of observed variables, exploiting an invariance property across permutations to identify the root cause.
**'Traversal'** [9, 24, 25] Requires the causal graph. Identifies nodes which are anomalous, have no anomalous parents, and are linked to the target node via a path of anomalous nodes as root causes.
Circa [47] Requires the causal graph. Fits a linear structural causal model to the data from the normal period and returns the node with the largest residual, given its parents, in the anomalous period.
**'Counterfactual'** [23] Requires the structural causal model. Finds the counterfactual Shapley contribution of each node to the anomalousness of the target event and outputs the node with the highest contribution.

We also evaluate against RCD [10] and $\varepsilon$-Diagnosis [11] in Appendix F, however, neither method operates in the single anomalous sample setting, and so we do not recommend direct comparison.

**Synthetic data generation:** full details are provided in Appendix F. In brief, to generate data, we sample a random DAG (not restricted to a polytree) of $n$ nodes. Structural equations are randomly assigned as simple feed-forward neural networks or as linear models of the parents and noise variables. In all experiments, 1000 observations are drawn according to the randomly assigned model in the normal period. To produce an anomalous sample, a root cause is chosen at random from among the nodes and a target node from among its descendants (including itself). An anomaly is then injected at the root cause by adding $x$ multiples of its standard deviation to its value, and propagated through the causal model. In each experiment, this process is repeated 100 times per data point. Unless otherwise stated, each algorithm is evaluated by its top-1 recall. With synthetic data, we answer the following questions:

**How does performance vary with anomaly strength?** We display results for $x \in \{2.0, 2.1, 2.2, \ldots, 3.0\}$ in Fig. 1. We find that the strongest performing algorithms are SMOOTH TRAVERSAL, Traversal, and Counterfactual, all of which outperform SCORE ORDERING. Circa and Cholesky perform considerably worse than the other approaches, apparently due to the assumption of linearity (see Fig. 5 in Appendix F). The performance of SCORE ORDERING improves slightly as the anomaly strength increases, but performance is approximately constant for the other approaches. We extend this analysis in Appendix F.2: to RCA and $\varepsilon$-Diagnosis (Fig. 4), using only linear SCMs (Fig. 5), and restricting the DAG to being a polytree (Fig. 6).

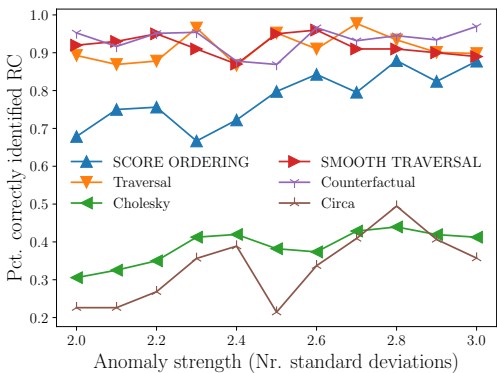

Figure 1: True positive rate for identifying the root cause against anomaly strength injected at the root cause.

**How do runtimes scale with increasing graph size?** For a fixed $x = 3$, we measure the runtime of each algorithm for graphs with an increasing number of nodes, generated as described above. We observe that (see Fig. 3 in Appendix F), Traversal and SMOOTH TRAVERSAL are the fastest, with the remaining approaches having comparable average runtimes.

**How does performance vary with increasing graph size?** We generate causal models with $n \in \{20, 40, \ldots, 100\}$ and fixed anomaly strength of $x = 3$ (see Fig. 7). While the performance of SMOOTH TRAVERSAL, Traversal and Counterfactual seems to remain constant, the performances of SCORE ORDERING, Cholesky, and Circa all begin to decrease as the graph increases in size.

**How robust are the methods to misspecification of the causal graph?** For the approaches that require a causal graph, we investigated how their performances varied as an increasing number of random edge additions/removals/reversals were introduced to the given graph (see Appendix F.4). For $n = 50$ and $x = 3.0$, we see SMOOTH TRAVERSAL, Traversal and Counterfactual are similarly robust, showing similar delays in performance, while the performance of Circa decayed more rapidly.

**Real world data evaluation:** We additionally performed a comprehensive evaluation with two 'real-world' datasets: the PetShop [27] dataset (see Appendix F.6) and the Sock-shop 2 dataset [26] (see Appendix F.7), as well as 'semi-synthetic' datasets generated using the ProRCA package [48] (see Appendix F.8). In these evaluations, the picture can be quite different from that in the synthetic data experiments. SCORE ORDERING often performs well across the datasets, but both of our algorithms are variously outperformed by Cholesky, Circa or Counterfactual in certain settings.

**Code**: https://github.com/amazon-science/RCAWithMissingStructuralKnowledgeCode

## 5   Conclusion

We have explored several directions in which the practical limitations of RCA can be addressed: first by avoiding estimation of conditional distributions in the setting that the causal graph is known, and second by avoiding needing the causal graph at all, in the setting that it is a polytree. To do so, we leverage information-theoretic (IT) anomaly scores and prove general laws about their typical decay along causal paths without any parametric assumptions. Using our results, we then propose two novel algorithms: SMOOTH TRAVERSAL and SCORE ORDERING, give probabilistic guarantees for the correctness of their outputs, and show they have competitive performance in both synthetic and real-world data despite their simplicity. To our knowledge, our work is the only which provides non-parametric guarantees for RCA with a single anomalous point without requiring the full structural causal model.

## Acknowledgments and Disclosure of Funding

We would like to thank Manuel Gomez-Rodriguez for fruitful discussions during different stages of this project. W. R. Orchard would like to thank Peterhouse, Cambridge, and Cancer Research UK for their support for this work.

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

## A  Relationship to contribution analysis

We first sketch the causal framework adopted by Budhathoki et al. [23]. In particular, they assume that the causal relationships between $X_1, \ldots, X_n$ are described by a Structural Causal Model (SCM). In an SCM, each variable $X_i$ is a function $f_i$ of its parents $PA_i$ in the causal graph, and an unobserved noise term $N_i$:

$$X_i := f_i(PA_i, N_i), \tag{14}$$

where the noise variables $N_1, \ldots, N_n$ are jointly independent [33]. In addition, it is assumed that the SCM is invertible [49]. That is, the noise value $n_j$ of $N_j$ can be recovered from the value $x_j$ of its corresponding observed variable $X_j$ and the values $pa_j$ of its parents.

As in our case, the value $x_n$ is flagged as an anomaly, and we wish to identify its root cause from among the variables $(X_1, \ldots, X_n)$. Treating (without loss of generality) $X_n$ as a sink node in the causal DAG, iterative application of Eq. 14 results in the representation

$$X_n = F(N_1, \ldots, N_n), \tag{15}$$

in which the structural information is implicit in the function $F$. By Eq. 15 we therefore have that $x_n = F(n_1, \ldots, n_n)$ where $(n_1, \ldots, n_n) := \mathbf{n}$ denote the corresponding values of the noise variables for sample $(x_1, \ldots, x_n)$. This representation makes clear that a node $X_j$'s contribution to the anomaly $x_n$ is ultimately attributable to the contribution of its noise term [23, 50].

If you consider our presentation of RCA, wherein we attribute the anomaly $x_n$ to one of the causal mechanisms being corrupted, the characterisation in terms of noise terms may appear quite different. However, we can connect the two perspectives by noting that one can think of each value $n_j$ as randomly switching between deterministic mechanisms $f_j(\cdot, n_j)$ – so-called *response functions* [51]. If a noise term is "corrupted" and takes an unusual value $n_j$, the corresponding mechanism generating $x_j$ from its parents $pa_j$ will be unusual as well. The goal of finding which causal mechanism did not work as expected, then is common between the two perspectives. The intuition behind the approach of Budhathoki et al. is that if for the root cause $X_j$ the unusual value of its noise term $n_j$ were replaced by a "normal" one, then $x_n$ would change to a non-anomaly with high probability. This gives the basis for defining how much each variable contributed to the observed anomaly. We explain how this is formalised next.

### A.1  Contribution analysis

To compute the contribution of each noise $N_i$ to the anomaly $x_n$, Budhathoki et al. [23] measure how replacing the observed value $n_i$ (originating from a potentially corrupted mechanism) with a random "normal" value, sampled from its regular distribution $P(N_i)$, changes the likelihood of the anomaly event, $E = \{\tau(X_n) \geq \tau(x_n)\}$. Intuitively, this shows us the extent to which $n_i$ was responsible for the extremeness of $x_n$.

Note, however, that the influence of the value $n_i$ on the likelihood of the anomaly event will also depend on the values of the other noise variables. In order to assess the contribution of $n_i$ we therefore also need to consider how it depends on the context, i.e. how it changes depending on which other noise variables we similarly replace with random "normal" values. To formalise this, for any subset of the index set, $\mathcal{S} \subseteq \mathcal{U} := \{1, \ldots, n\}$, first note that the probability of the anomaly event when all nodes in $\mathcal{S}$ are set to their observed value and all the nodes in $\bar{\mathcal{S}} = \mathcal{U} \setminus \mathcal{S}$ are randomised is $P(E \mid \mathbf{N}_\mathcal{S} = \mathbf{n}_\mathcal{S})$. Now the contribution of a node $j \notin \mathcal{S}$ given the context $\mathcal{S}$ is defined as:[7]

$$C(j|\mathcal{S}) := \log \frac{P(E|\mathbf{N}_\mathcal{S} = \mathbf{n}_\mathcal{S}, \mathbf{N}_j = \mathbf{n}_j)}{P(E|\mathbf{N}_\mathcal{S} = \mathbf{n}_\mathcal{S})}. \tag{16}$$

To give a "fair" attribution among the noise variables, the authors adopt Shapley values. Let $\Pi : \mathcal{U} \to \mathcal{U}$ be the set of all possible permutations of the nodes and $\pi \in \Pi$ be any permutation. One then defines the contribution of a node $j$ given permutation $\pi$ as $C^\pi(j) := C(j|I^{\pi<j})$, where $I^{\pi<j}$ denotes the set of nodes that appear *before* $j$ with respect to $\pi$, *i.e.*, $I^{\pi<j} = \{i \in \mathcal{U} \mid \pi(i) < \pi(j)\}$. We easily see that for each permutation $\pi$, $S(x_n)$ decomposes into the contributions of each node,

---

[7]Note the difference in the definition of $C(.|\mathcal{S})$ compared to [23]. In our case $\mathcal{S}$ is the set for which $\mathbf{N}_\mathcal{S} = \mathbf{n}_\mathcal{S}$ while in [23] $\mathbf{N}_{\mathcal{U}\setminus\mathcal{S}} = \mathbf{n}_{\mathcal{U}\setminus\mathcal{S}}$.

*i.e.*, $S(x_n) = \sum_{j \in \{1,...,n\}} C^{\pi}(j)$. The Shapley contribution of a node is then given by averaging over all the possible permutations in $\Pi$:

$$C^{Sh}(j) := \frac{1}{n!} \sum_{\pi \in \Pi} C(j|I^{\pi<j}). \tag{17}$$

This approach therefore provides a full quantitative contribution analysis of root causes. However, it suffers from some practical issues. Firstly, the Shapley contribution of a node is expensive to compute [23]. More fundamentally, however, for most permutations $\pi$, the contributions $C^{\pi}(j)$ rely on knowing the structural equations (14). As the SCM cannot generally be inferred even with interventional data, this is a serious bottleneck for the application of this approach. As the approaches we propose in this paper do not depend on knowing the structural equations (or even the causal graph in the case of SCORE ORDERING), we explain next how we can nonetheless recast our results in terms of the contribution approach just presented.

## A.2 Interventional vs counterfactual RCA

The key result we will need is that so long as $\pi$ is a topological ordering of the nodes in the causal graph, then all contributions $C^{\pi}(j)$ do not require knowing the SCM. To illustrate this, first consider the bivariate case $X \to Y$ with structural equations $X := N_X$ and $Y := f(X, N_Y)$, and anomaly $y$ in sample $(x, y)$. To determine the contribution of $N_Y$ to the anomaly, we consider randomising $N_Y$ and fixing $N_X = n_X = x$. This generates $Y$ according to the observational distribution $P(Y \mid x)$, so its estimation does not require knowledge of the SCM. On the other hand, randomising $N_X$ and fixing $N_Y = n_Y$ cannot be resolved into any observational term (see also Section 5 in [52]).[8]

The following result generalises this insight:

**Proposition A.1.** *Whenever $\pi$ is a topological order of the causal DAG, i.e., there are no arrows $X_i \to X_j$ for $\pi(i) > \pi(j)$, all contributions $C^{\pi}(j)$ can be computed from observational conditional distributions.*

This will allow us to connect our approach to that of Budhathoki et al. [23] in the following sense: if our goal is to simply identify a single root cause, then as long as it is "dominating" in the sense that it has the largest contribution of any variable regardless of the order chosen, Proposition A.1 says we can identify it without knowing the SCM.

It will prove instructive to generalise the notion of the contribution of a *single node* in (16) to the contribution of a *set* $\mathcal{R} \subseteq \mathcal{U} \setminus \mathcal{S}$ of nodes, given the context $\mathcal{S}$, *i.e.*,

$$C(\mathcal{R}|\mathcal{S}) := \log \frac{P(E|\mathbf{N}_{\mathcal{R}} = \mathbf{n}_{\mathcal{R}}, \mathbf{N}_{\mathcal{S}} = \mathbf{n}_{\mathcal{S}})}{P(E|\mathbf{N}_{\mathcal{S}} = \mathbf{n}_{\mathcal{S}})}. \tag{18}$$

This notion becomes rather intuitive after observing that it is given by the sum of the contributions of all the elements in $\mathcal{R}$ when they are one by one added to the context $\mathcal{S}$:

**Lemma A.2.** *For any set $\mathcal{R} \subseteq \mathcal{U} \setminus \mathcal{S}$, it holds that $C(\mathcal{R}|\mathcal{S}) := C(j_1|\mathcal{S}) + \sum_{i=2}^{k} C(j_i|\mathcal{S} \cup \{j_1, \ldots, j_{i-1}\})$ with $\mathcal{R} = \{j_1, \ldots, j_k\}$.*

Next, the following result shows that a set of nodes are unlikely to obtain a high contribution when the noise values are randomly drawn from their usual distributions, *i.e.*, we have the following proposition:

**Proposition A.3.** *Whenever all noise variable in some set $\mathcal{R}$ are sampled from $P(\mathbf{N}_{\mathcal{R}})$, it holds that $P(C(\mathcal{R}|\mathcal{S}) \geq \alpha) \leq e^{-\alpha}$.*

Note that the proposition allows that the variables $N_j$ *not* in $\mathcal{R}$ are drawn from a different distribution. Thus, Proposition A.3 states that it is unlikely that a set that only contains *non-corrupted nodes* has high contribution.

Next, we will show how our main results can be recast in terms of bounds on contributions, rather than on $p$-values.

---

[8]This can be checked by an SCM with two binaries where $Y := X \oplus N_Y$, with unbiased $N_Y$, which induces the same distribution as the SCM $Y := N_Y$, where $X$ and $Y$ are disconnected and thus $X$ has a contribution of zero.

### A.3 Bounds on contributions

Consider, for illustration, the setting discussed in subsection 3.1 wherein the causal DAG is $X \to Y$, we observe $(x, y)$ with anomaly scores $S(x), S(y)$, and we wish to find the root cause of anomaly $y$. Under the same assumptions as stated there we have the following proposition:

**Proposition A.4.** *Subject to score typicality, the contributions of $y$ and $x$ on the anomaly $y$ satisfy:*

$$C(x) \leq S(x) \quad and \quad C(y \mid x) \geq |S(y) - S(x)|_+$$

*Proof.* Proposition A.1 permits rewriting contributions in terms of observational probabilities, so that, together with score typicality, we have:

$$
\begin{aligned}
C(y \mid x) :&= \log \frac{P(\tau(Y) \geq \tau(y) \mid Y = y, X = x)}{P(\tau(Y) \geq \tau(y) \mid X = x)} \\
&= \log \frac{1}{P(\tau(Y) \geq \tau(y) \mid X = x)} \\
&= S(y \mid x) \geq |S(y) - S(x)|_+,
\end{aligned}
$$

and using $C(x) + C(y \mid x) = S(y)$ we obtain the bound for $C(x)$. $\qquad\square$

We can now further interpret the conclusions drawn in subsection 3.1, in terms of contributions. For example, suppose we have $S(y) \gg S(x)$. In the case that $X \to Y$, Proposition A.4 says that $y$ must have a large contribution to the anomaly, while $x$ must have a smaller one. We would therefore favour $y$ as the root cause. Alternatively, if $S(x) \gg S(y)$, then we cannot conclude that $y$ has a large contribution, but we would reject the hypothesis that the mechanism generating $x$ worked as normal (from Lemma 3.2), and its contribution may be large. Under the working hypothesis that there is only a single root cause, we would favour $x$. Following the extensions from the bivariate case in subsections 3.2 and 3.3, we can similarly recast the results in terms of contributions as above. Naturally, this does not alter any of the conclusions, but rather demonstrates that our results do not depend strictly on our adopted formalisation of the RCA problem in terms of CBNs, but additionally admits an interpretation in terms of contributions.

## B  Sample complexity of information-theoretic anomaly score estimation

To reliably tell whether the IT anomaly score of one anomaly exceeds that of another, we need to estimate each score up to an error of at least half their difference. We investigate how many samples are required to estimate the scores $S(x_i), \ldots, S(x_n)$ up to an error level $\delta$:

**Lemma B.1.** *If we use at least*

$$k = \frac{3e^{S_{\max}}}{\delta^2} \log\left(\frac{2n}{\alpha}\right) \tag{19}$$

*samples from the 'normal' period, the score estimates $\hat{S}(x_1), \ldots, \hat{S}(x_n)$ using the estimator in Equation (5) satisfy $|\hat{S}(x_i) - S(x_i)| < \delta$ for all $i \in \{1, \ldots, n\}$ with probability at least $1 - \alpha$.*

*Proof.* Let $p_i := P(\tau(X_i) \geq \tau(x_i))$ and estimate $\hat{p}_i := \frac{1}{k} \sum_{j=1}^{k} \mathbf{1}\{\tau(x_i^j) \geq \tau(x_i)\}$ so that $S(x_i) = -\log p_i$ and $\hat{S}(x_i) = -\log \hat{p}_i$ as in Equations (4) and (5), respectively. $|\hat{S}(x_i) - S(x_i)| < \delta$ is equivalent to,

$$|\log \hat{p}_i - \log p_i| < \delta \implies e^{-\delta} < \frac{\hat{p}_i}{p_i} < e^{\delta}. \tag{20}$$

Rearranging, we have $p_i(e^{-\delta} - 1) < \hat{p}_i - p_i < p_i(e^{\delta} - 1)$, and noting $e^{\delta} - 1 > 1 - e^{-\delta}$ for all $\delta > 0$, we have

$$|\hat{p}_i - p_i| < \epsilon_i := (e^{\delta} - 1)p_i, \tag{21}$$

so that when $\delta$ is small we have $\epsilon_i \approx \delta p_i$. We can then use the multiplicative form of the Chernoff bound for $0 \leq \delta \leq 1$ to say,

$$P(|\hat{p}_i - p_i| \geq \delta p_i) \leq 2\exp\left(-\frac{\delta^2 p_i k}{3}\right). \tag{22}$$

Noting that $\min_i p_i = \min_i e^{-S(x_i)} = e^{-\max_i S(x_i)} =: e^{-S_{\max}}$ and applying the union bound,

$$P\left(\bigcup_{i=1}^{n}\{|\hat{p}_i - p_i| \geq \delta p_i\}\right) \leq 2n \exp\left(-\frac{\delta^2 e^{-S_{\max}} k}{3}\right), \tag{23}$$

so that finally,

$$2n \exp\left(-\frac{\delta^2 e^{-S_{\max}} k}{3}\right) < \alpha \quad \Longrightarrow \quad k > \frac{3 e^{S_{\max}}}{\delta^2} \log\left(\frac{2n}{\alpha}\right)$$

$\square$

## C  Proofs

### C.1  Proof of Lemma 3.4

Since we assume that $S(X)$ is a continuous variable with density with respect to the Lebesgue measure, it follows from the properties of IT scores that it is distributed according to the density $p(S(X) = s) = e^{-s}$ for $s \geq 0$. Accordingly, the conditional distribution of $S(X)$, given $S(X) \in [s_X, S(y)]$ has the density $p(S(X) = s \mid S(X) \in [s_X, S(y)]) = e^{-s}/(e^{s_X} - e^{-S(y)})$. We want to compare averages of the expressions

$$P(S(Y) \geq S(y)|S(X) = s) \quad \text{versus} \quad e^{-|S(y)-s|_+}$$

over the interval $[s_X, S(y)]$. Averaging the difference of the two expressions with respect to the above-mentioned density yields

$$\int_{s_X}^{S(y)} \left(P(S(Y) \geq S(y)|S(X) = s) - e^{-|S(y)-s|_+}\right) p(s \mid S(y) \geq S(X) \geq s_X) ds$$

$$= P(S(Y) \geq S(y)|S(y) \geq S(X) \geq s_X) - \int_{s_X}^{S(y)} e^{s-S(y)} \frac{e^{-s}}{e^{-s_X} - e^{-S(y)}} ds$$

$$\leq \frac{e^{-S(y)}}{e^{-s_X} - e^{-S(y)}} - \int_{\tilde{s}_X}^{S(y)} \frac{e^{-S(y)}}{e^{-\tilde{s}_X} - e^{-S(y)}} ds = \frac{e^{-S(y)}}{e^{-s_X} - e^{-S(x)}}(1 - S(y) + s_X) \leq 0.$$

The first inequality holds because $P(B|A) \leq P(B)/P(A)$ for any two events $A, B$ and the second because we have assumed $s_X \leq S(y) - 1$.

Applying Markov's inequality, we therefore have

$$P_{X|S(X) \in [s_X, S(y)]}[P(S(Y) \geq S(y) \mid S(X)) \geq c \cdot e^{-|S(y)-S(X)|_+}] \leq 1/c. \tag{24}$$

As $X$ may in general be multivariate, $S$ will not in general be injective, and so we cannot substitute conditioning on a particular value of the score for conditioning on the associated $x$, which is required to give a bound on the conditional score. As such, note that

$$P(S(Y) \geq S(y) \mid S(X) = s) = \int P(S(Y) \geq S(y) \mid X = x) \, p(x \mid S(X) = s) \, dx,$$

so that applying Markov's inequality for any given $s$,

$$P_{X|S(X)=s}[P(S(Y) \geq S(y) \mid X) \geq c \cdot P(S(Y) \geq S(y) \mid S(X))] \leq 1/c. \tag{25}$$

As the inequality holds for all $s$, we can take the average of the LHS with respect to $p(S(X) = s \mid S(X) \in [s_X, S(y)])$ to say,

$$P_{X|S(X) \in [s_X, S(y)]}[P(S(Y) \geq S(y) \mid X) \geq c \cdot P(S(Y) \geq S(y) \mid S(X))] \leq 1/c. \tag{26}$$

Applying the union bound to the events in equations 24 and 26, the joint event

$$P(S(Y) \geq S(y) \mid S(X)) \leq c \cdot e^{-|S(y)-S(X)|_+} \quad \text{and}$$

$$P(S(Y) \geq S(y) \mid X) \leq c \cdot P(S(Y) \geq S(y) \mid S(X))$$

occurs with probability at least $1 - 2/c$, and in such case the following inequality holds:

$$P(S(Y) \geq S(y) \mid X = x) \leq c^2 \cdot e^{-|S(y)-S(x)|_+}. \tag{27}$$

Finally, noting we can exchange $S(Y) \geq S(y)$ for $\tau(Y) \geq \tau(y)$ due to the properties of IT scores, and taking the negative logarithm of both sides, for $x$ sampled at random according to $P(X = x \mid S(X) \in [s_X, S(y)])$,

$$S(y \mid x) \geq |S(y) - S(x)|_+ - 2 \log c, \tag{28}$$

holds with probability at least $1 - 2/c$, as desired.

## C.2 Proof of Lemma 3.5

$$P(\tau(Y) \geq \tau(y) \mid X = x) = P(S(Y) \geq S(y) \mid X = x) = P(S(Y) \geq S(y) \mid S(X) = S(x))$$

$$\leq P(S(Y) \geq S(y) \mid S(X) \geq S(x)) \leq \frac{P(S(Y) \geq S(y))}{P(S(X) \geq S(x))} = \frac{e^{-S(y)}}{e^{-S(x)}}.$$

Where the first equality follows from the definition of IT scores and the second from injectivity. The first inequality follows from monotonicity, and the second because $P(B \mid A) \leq P(B)/P(A)$ for any two events $A, B$. The final equality again follows from the definition of IT scores. Taking the negative logarithm of both sides, and remembering that anomaly scores are non-negative, we prove the lemma.

## C.3 Proof of Lemma 3.7

Assume without loss of generality that $X_1, \ldots, X_n$ is a topological ordering of the DAG. Then $S(X_i|X_1, \ldots, X_{i-1}) = S(X_i|PA_i)$ holds due to the local Markov condition. Since $S(X_j|PA_j = pa_j)$ has density $e^{-s}$ for all $pa_j$, also $S(X_i|X_1 = x_1, \ldots, X_{i-1} = x_{i-1})$ has density $e^{-s}$ for all $x_1, \ldots, x_{i-1}$. Hence, $S(X_i|PA_i)$ is independent of $X_1, \ldots, X_{i-1}$.

## C.4 Proof of Lemma 3.8

From Lemma 3.7, and the properties of IT scores, if $H_0^j$ holds then $S(X_i \mid PA_i)$ for $i \neq j$ are independent, standard Exponential random variables. The sum of $n - 1$ independent, standard Exponential random variables is Erlang distributed [53] with CDF

$$F(x) = 1 - e^{-x} \sum_{l=0}^{n-2} \frac{x^l}{l!},$$

and as such, $S_{\text{sum}}$ is Erlang distributed. By the probability integral transform $U := F(S_{\text{sum}})$ is uniformly distributed on $[0, 1]$, and so by symmetry, so is $1 - U$. Again by the probability integral transform, $-\log(1 - U)$ is standard Exponential, so finally noting that $S = -\log(1 - F(S_{\text{sum}}))$, we have the result.

## C.5 Proof of Theorem 3.9

The theorem is a direct result of Lemma 3.7 and Lemma 3.8.

## C.6 Proof of Theorem 3.11

$$s_{\text{sum}} := \sum_{i \neq j} S(x_i|pa_i) \geq \sum_{i \neq j} |S(x_i) - S(pa_i)|_+ =: \hat{s}_{\text{sum}}, \tag{29}$$

where the inequality follows from score typicality of event $(x_1, \ldots, x_n)$, and $s_{\text{sum}}$ and $\hat{s}_{\text{sum}}$ are the realisations of $S_{\text{sum}}$ and $\hat{S}_{\text{sum}}$, respectively. The theorem then follows directly from Theorem 3.9 with $\hat{S}_{\text{sum}}$ in place of $S_{\text{sum}}$.

## C.7 Proof of Theorem 3.12

From Lemma 3.7 we have that the conditional anomaly score for the anomaly with the maximum score gap must be at least $\delta_{\max}$. From Lemma 3.4, and the properties of IT anomaly scores, the conditional anomaly scores are independent and each distributed according to density $p(s) = e^{-s}$. Under $H_0^{\max}$, assume without loss of generality that $x_n$ corresponds to the true root cause. Then the remaining $(n - 1)$ conditional scores are i.i.d. $\text{Exp}(1)$, and

$$P\big(S(X_i \mid \text{PA}_i) < \delta \text{ for all } i < n\big) = \big(P(S(X_i \mid \text{PA}_i) < \delta)\big)^{n-1} = (1 - e^{-\delta})^{n-1}.$$

Therefore, the probability of observing at least one conditional anomaly score of at least $\delta_{\max}$ among the $(n - 1)$ non-root causes is $1 - (1 - e^{-\delta_{\max}})^{n-1}$. This probability upper bounds the probability of observing at least one score gap of $\delta_{\max}$ or greater as score gaps lower bound conditional scores.

## C.8 Proof of Theorem 3.13

The proof of the theorem is already almost complete, following the argument in the main text. As noted there, $H_0^{\text{top-}k}$ implies that along the causal path the score of $x_{\pi(1)}$ must be higher than an ancestral node by at least $\Delta_k$. Lemma 3.3 along with the coarsening argument, implies that, individually, the hypothesis that the mechanism generating $x$ worked as expected, where $S(x)$ is higher than the score of an ancestral node by $\Delta_k$, can be rejected at level $p \leq e^{-\Delta_k}$. Therefore, along the causal path, the probability that the score of *any* of the (maximum of) $n$ nodes exceeds an ancestral node by $\Delta_k$ is at most $n \cdot e^{-\Delta_k}$, applying the union bound. However, note that we need to reapply this argument for each of the chains incident on $X_{\pi(1)}$ in the graph. The number of incident chains is at most $d_{\max}$, and so again applying the union bound, we arrive at the final bound given in the theorem.

## C.9 Proof of Proposition A.1

With the event $E$ as in Eq. 3 and $C^\pi(j) := C(j|I^{\pi<j})$ we have

$$
C^\pi(j) = \log \frac{P(E|\mathbf{N}_{\pi<j \,\cup\{j\}} = \mathbf{n}_{\pi<j \,\cup\{j\}})}{P(E|\mathbf{N}_{\pi<j} = \mathbf{n}_{\pi<j})} \overset{i}{=} \log \frac{P(E|\mathbf{X}_{\pi<j \,\cup\{j\}} = \mathbf{x}_{\pi<j \,\cup\{j\}})}{P(E|\mathbf{X}_{\pi<j} = \mathbf{x}_{\pi<j})}
$$

$$
\overset{ii}{=} \log \frac{P(E|do(\mathbf{X}_{\pi<j \,\cup\{j\}} = \mathbf{x}_{\pi<j \,\cup\{j\}})}{P(E|do(\mathbf{X}_{\pi<j} = \mathbf{x}_{\pi<j}))}. \tag{30}
$$

(i) and (ii) are seen as follows:

$$
P(E|\mathbf{N}_\mathcal{I}) = P(E|\mathbf{X}_\mathcal{I})) = P(E|do(\mathbf{X}_\mathcal{I})). \tag{31}
$$

The first equality in Eq. 31 follows from $X_n \perp X_\mathcal{I}\,|\mathbf{N}_\mathcal{I}$ and because $\mathbf{X}_\mathcal{I}$ is a function of $\mathbf{N}_\mathcal{I}$. The second one follows because conditioning on all ancestors blocks all backdoor paths. Note that since $\pi$ is a topological ordering of the nodes, all $\pi < j$ are ancestors of $j$.

## C.10 Proof of Lemma A.2

Denote $q(\mathcal{S}) = P(E\,|\,\mathbf{N}_\mathcal{S} = \mathbf{n}_\mathcal{S})$ then we have that

$$
\begin{aligned}
C(\mathcal{R}|\mathcal{S}) &= \log q(\mathcal{S} \cup \mathcal{R}) - \log q(\mathcal{S}) \\
&= (\log q(\mathcal{S} \cup \mathcal{R}) - \log q(\mathcal{S} \cup \mathcal{R} \setminus \{j_k\})) + \ldots \\
&\quad (\log q(\mathcal{S} \cup \mathcal{R} \setminus \{j_k\}) - \log q(\mathcal{S} \cup \mathcal{R} \setminus \{j_k, j_{k-1}\})) + \ldots \\
&\quad + (\log q(\mathcal{S} \cup \{j_1\}) - \log q(\mathcal{S})) = C(j_1|\mathcal{S}) + \sum_{i=2}^{k} C(j_i|\mathcal{S} \cup \{j_1, \ldots, j_{i-1}\}).
\end{aligned}
$$

## C.11 Proof of Proposition A.3

By definition, we have that

$$
C(\mathcal{R}|\mathcal{S}) = \log \frac{P(E|\mathbf{N}_R = \mathbf{n}_R, \mathbf{N}_S = \mathbf{n}_S)}{P(E|\mathbf{N}_S = \mathbf{n}_S)},
$$

and we use $P(E|\mathbf{n}_S)$ instead of $P(E|\mathbf{N}_S = \mathbf{n}_S)$ when it is clear from the context.

Further, $C(\mathcal{R}|\mathcal{S})$ is actually a function of $\mathbf{n}_R$ and $\mathbf{n}_S$. For fixed $\mathbf{n}_S$, define the set $B := \{\mathbf{n}_R\,|C(\mathcal{R}|\mathcal{S}) \geq \alpha\}$. It can equivalently be described by

$$
B = \{\mathbf{n}_R\,|\log \frac{P(E|\mathbf{n}_R, \mathbf{n}_S)}{P(E|\mathbf{n}_S)} \geq \alpha\} = \{\mathbf{n}_R\,|P(E|\mathbf{n}_R, \mathbf{n}_S) \geq P(E|\mathbf{n}_S) \cdot e^\alpha\}.
$$

We thus have

$$
P(E|\mathbf{n}_R \in B, \mathbf{n}_S) \geq P(E|\mathbf{n}_S) \cdot e^\alpha.
$$

Hence,

$$
\frac{P(E, \mathbf{n}_R \in B|\mathbf{n}_S)}{P(\mathbf{n}_R \in B|\mathbf{n}_S)} \geq P(E|\mathbf{n}_S) \cdot e^\alpha.
$$

Using

$$P(E|\mathbf{n}_S) \geq P(E, \mathbf{n}_R \in B|\mathbf{n}_S)$$

we obtain

$$P(\mathbf{n}_R \in B|\mathbf{n}_S) \leq e^{-\alpha}.$$

## D Visualising polytrees

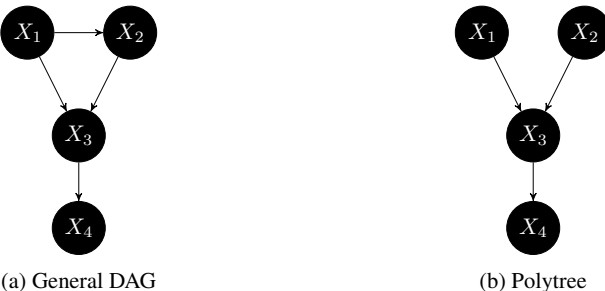

(a) General DAG          (b) Polytree

Figure 2: (a) A general DAG may contain undirected cycles, as between $X_1 - X_2 - X_3 - X_1$. (b) A polytree cannot contain undirected cycles so could not have an edge between $X_1$ and $X_2$ without first removing another.

## E Is increase of scores along causal pathways rare?

Li et al. [46] describe a condition under which the $z^2$-score of the effect $X_j$ can be larger than the $z^2$-score from the cause $X_i$ in a DAG with linear structural equations. Their Theorem 2.1 states that this happens (in the limit of large perturbations, see the details further below) if and only if the variances $\sigma_i^2, \sigma_j^2$ of $X_i, X_j$ satisfy

$$\sigma_j^2 < \alpha_{i \rightarrow j}^2 \sigma_i^2, \tag{32}$$

where $\alpha_{i \rightarrow j}^2$ denotes the structural coefficient for the effect from $X_i$ to $X_j$ (which consists of the sum over the product of all direct influences along the paths from $i$ to $j$). Although our results refer to IT scores instead of $z^2$-scores, it is at the same time an example for increasing IT scores since the $z^2$-score can be turned into an IT score by a monotonic recalibration. It is therefore important that we understand this result if we are to understand if we expect SCORE ORDERING to be applicable in cases beyond polytrees. For an intuitive understanding we should first note that (32) describes the regime of *strong negative confounding*. This is because an unconfounded causal influence $X_j = \alpha_{i \rightarrow j} X_i + E_j$ with independent error term $E_j$ results in $\sigma_j^2 = \alpha_{i \rightarrow j}^2 \sigma_i^2 + \text{Var}(E_j)$. We are therefore in the regime where the noise is so strongly negatively correlated with the cause that it decreases the variance instead of increasing it.

Let us now discuss whether this phenomenon is rare or not when $X_i$ and $X_j$ are variables in a larger network (note that the following working is strongly inspired by [46]). To this end let $X_1, \ldots, X_n$ be causally ordered with structural equations

$$\mathbf{X} = A\mathbf{X} + \mathbf{N}, \tag{33}$$

where $A$ is a strictly lower triangular matrix and $\mathbf{N}$ is the vector of independent noise variables. To simplify notation, let all variables have zero mean.

The covariance matrix $\Sigma_{\mathbf{XX}}$ can be derived by well-known algebra:

$$\Sigma_{\mathbf{XX}} = (I - A)^{-1} \Sigma_{\mathbf{NN}} (I - A)^{-T},$$

where $\Sigma_{\mathbf{NN}}$ denotes the covariance matrix of the noise, which is diagonal by assumption. Defining

$$L := (I - A)^{-1} \Sigma_{\mathbf{NN}}^{1/2},$$

we obtain the unique Cholesky decomposition of $\Sigma_{\mathbf{XX}}$ since $\Sigma_{\mathbf{XX}} = LL^T$, where $L$ is lower diagonal (with non-zero diagonal by construction if we assume non-zero noise variance). The matrix $L$ has

interesting interpretations:

(i) $L$ generates the observations from independent noise variables as sources, formally $\mathbf{X} = L\tilde{\mathbf{N}}$, where $\tilde{\mathbf{N}} := \Sigma_{\mathbf{NN}}^{-1/2}\mathbf{N}$ is the standardized version of the noise from the original structural equation (33).

(ii) The squared row sums of $L$ thus show the variance of the observed variables, that is, $\sum_j L_{ij}^2 = \sigma_i^2$. Henceforth, we will assume it to be 1 without loss of generality.

(iii) The $i$th column of $L$ describes how shifting the noise $\tilde{n}_i$ to $\tilde{n}_i + 1$ changes the observations $\mathbf{x}$, that is, the observation $\mathbf{x} = L\tilde{\mathbf{n}}$ is changed to $\mathbf{x} + Le_i$, where $e_i$ denotes the $i$th canonical basis vector.

Since all the $X_j$ are standardized and centered, their $z^2$ score is simply their squared value $x_j^2$. If we shift the noise $\tilde{n}_i$ by a large value, the $z^2$-scores $x_i^2$ of the root cause $X_i$ versus the score $x_j^2$ of the effect $X_j$ satisfies the ratio $L_{ii}^2/L_{ij}^2$ in the limit of infinitely strong perturbation. The question of whether scores of effects are typically smaller than scores of their root causes now boils down to the following question: Given a lower triangular matrix $L$ whose row vectors have norm 1. Is $L_{ij}^2$ "typically" smaller than $L_{ii}^2$? We will discuss this question for the assumption that $n$ is large and that $\rho := \min_i\{L_{ii}^2\}$ is much larger than $1/n$. This assumption simply formalizes the idea that no $X_i$ is close to be a deterministic function of its ancestors. It may seem strong, but we could easily extend the below discussion to the case where only *most* of the variables satisfy this condition.[9] We conclude:

$$\sum_{1 \leq i < j} L_{ij}^2 \leq 1 - \rho.$$

Hence, the number $k$ of $i$ with $i < j$ for which $L_{ii}^2 < L_{ij}^2$ is at most $(1 - \rho)/\rho$. Using $\rho \gg 1/n$ we conclude $k \ll n$. Thus, only for a small fraction $k/n$ of root causes $i$, the score $L_{ii}^2$ of the root cause is smaller than the score $L_{ij}^2$ of the downstream effect $j$. In this sense, we may consider effects with larger score than the root cause as a rare phenomenon, even for *fixed* causal structural equations, not only as a statistical statement over multiple randomly generated structures.

# F Experimental details and further experiments

## F.1 Experimental details

To run Circa and Counterfactual, we used the implementations available in [27] using default parameters. We wrote our own implementation of Traversal (with anomaly score threshold of 3), and our own algorithms, SMOOTH TRAVERSAL AND SCORE ORDERING. The code for Cholesky is available at [46], however, there are in fact three different algorithms. For all experiments involving real-world data (see subsection F.6 below) we applied their "main" algorithm, using default parameters. For the synthetic experiments, we had to opt for the "highdim" version as the alternatives had already become prohibitively slow for the graph sizes considered. To generate an SCM for the experiments in Section. 4 (see Fig. 1), we first uniformly sample between 10 and 20 root nodes (20% to 40% of the total nodes of the graph) and uniformly assign to each either a standard Gaussian, uniform, or mixture of Gaussians as its noise distribution. As a second step, we recursively sample non-root nodes. Non-root nodes need not be sink nodes. The number of parent nodes that each non-root node is conditioned on is randomly chosen following a distribution that assigns a lower probability to a higher number of parents. In total, the causal graph is composed of 50 nodes. The parametric forms of the structural equations are randomly assigned to be either a simple feed-forward neural network with a probability of 0.8 (to account for non-linear models) and a linear model. The feed-forward neural network has three layers (input layer, hidden layer, and output layer) where the hidden layer has a number of nodes chosen randomly between 2 and 100. All the parameters of the neural network are sampled from a uniform distribution between -5 and 5. For the linear model, we sample the coefficients of the linear model from a uniform distribution between -1 and 1 and set the intercept to 0. In both cases, we use additive Gaussian noise as the relation between the noise and the variables.

---

[9]Note that *simulated* data can easily violate this assumption since many simulation schemes result in artifacts where variables that are late in the causal order can be almost perfectly reconstructed from others, a phenomenon called $R^2$-sortability in [54]

To generate data for the non-anomalous regime, we sample the noise of each of the variables and propagate the noise forward using the previously sampled structural equations. As mentioned in the main text, to produce anomalous data, we choose a root cause at random from the list of all nodes and a target node from the set of its descendants (including itself). Then we sample the noise of all variables and modify the noise of the root cause by adding $x$-times standard deviations (of its marginal distribution), where $x \in \{2, 2.1, 2.2, \ldots, 3\}$, and propagate the modified noise through the SCM to obtain a realisation of each variable. We repeat this process 100 times for each $x$ value added to the standard deviation and consider the algorithm successful if its result coincides with the chosen root cause.

**Computing Infrastructure.** All the experiments were run on a MacBook Pro with 16GB of memory with an Apple M1 processor.

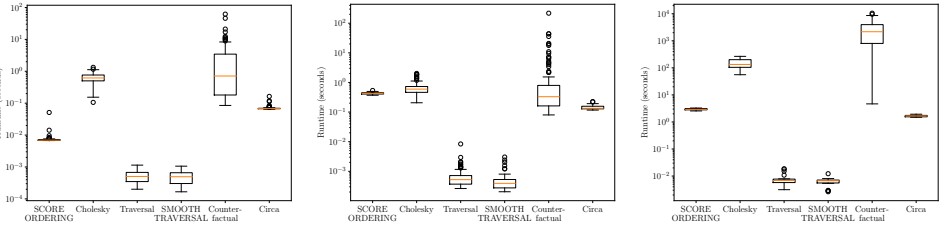

Figure 3: Runtimes of the algorithms for the experiment in Fig. 1; that is 50 nodes (left), an SCM with 100 nodes (center) and one with 1k nodes (right) (refer to the generation process above). The boxplots are produced using the default implementation in Matplotlib [55]. Note the log scale in the vertical axis.

## F.2 Further experiments varying root cause anomaly strength

**Methods which require multiple samples from the anomalous regime** Most RCA approaches require multiple samples from the anomalous regime. For completeness, we extended the evaluation in Figure 1 to include two such methods: RCD [29] and $\varepsilon$-Diagnosis [11], as neither require the causal graph (see Figure 4). For both methods we made use of the implementations available in the PyRCA library [56] using default parameters. While all other methods were provided with only a single sample from the anomalous regime, both RCD and $\varepsilon$-Diagnosis were provided with 100, and so we believe that direct comparison is not appropriate. The experiment was otherwise identical to that described above in Appendix F. Nonetheless, we see that while RCD has very strong performance, with a top-1 recall close to one at all anomaly strengths, $\varepsilon$-Diagnosis is the opposite, with a recall close to zero for all anomaly strengths. It is worth noting that the original authors of the RCD method [10] show in their supplemental material that performance is very poor in the low-sample regime (<100 anomalous samples, see Figure 2 therein). We conclude that RCD could be preferable in scenarios where one has access to many samples in the anomalous period, but that in the low-sample regime SCORE ORDERING is preferable.

**How do the algorithms perform when the structural causal model is linear?** To test whether the performance of Circa and Cholesky in Fig. 1 was poor compared to the other algorithms due to the assumption of linearity, we reproduced the experiment as detailed in Section 4 and Appendix F above, with the only change being that now all structural equations were sampled to be linear. Indeed, Fig. 5 shows that Circa now becomes one of the best performing algorithms, alongside SMOOTH TRAVERSAL, Traversal, and Counterfactual. Similarly, although Cholesky remains below the top performers, its performance is notably improved at all anomaly strengths, even now outperforming SCORE ORDERING.

**How do the algorithms perform when the causal graph is a polytree?** In all synthetic experiments, we placed no restrictions on the structure of the causal graph other than that it be a DAG. Our main theorems are stated, however, for polytrees. We therefore repeated the experiments from Figure 1 with the single change constraining the simulated causal graphs to be polytrees (see Figure 6). Unsurprisingly, methods which do not make use of the causal graph have unchanged performance, and and those that do are either unchanged or show improvement, including SMOOTH TRAVERSAL.

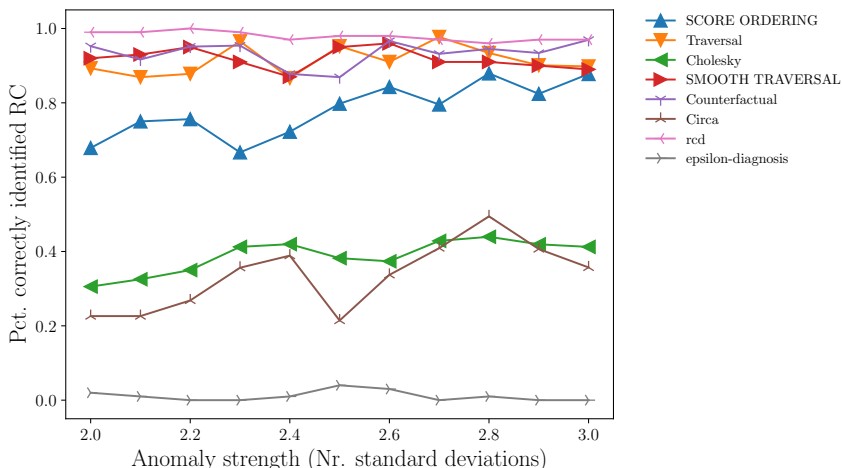

Figure 4: True positive rate for identifying the root cause against anomaly strength injected at the root cause, including RCD and $\varepsilon$-Diagnosis. Note that both RCD and $\varepsilon$-Diagnosis are given 100 samples from the anomalous period, while all other methods are given only one.

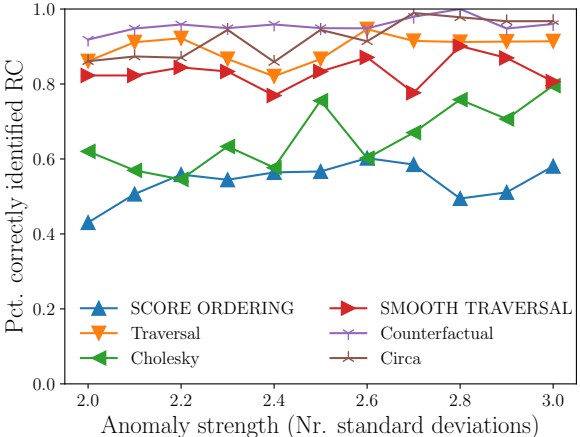

Figure 5: True positive rate for identifying the root cause against anomaly strength injected at the root cause, when all structural equations are linear.

Interestingly, Circa shows a marked improvement in performance despite not depending on the polytree assumption.

### F.3 Further experiments by varying the number of nodes in the graph

We also run experiments by fixing the number of standard deviations added ($x$ in Section. 4) to 3 and varying the number of nodes in the SCM. Here, we have chosen the numbers $\{20, 40, 60, 80, 100\}$. We observe in Fig. 7 that the performance for Traversal, SMOOTH TRAVERSAL, and Counterfactual does not change much for different graph sizes, whereas the performance of SCORE ORDERING, Cholesky and Circa decreases slightly for larger graph sizes.

### F.4 How does performance change when the given causal graph is misspecified

To mimic the scenario where one has access to an imperfect causal graph, we investigate how robust the performance each of the algorithms (which require a causal graph) is to its misspecification (8). The data were generated as described at the beginning of Appendix F, but with a fixed anomaly

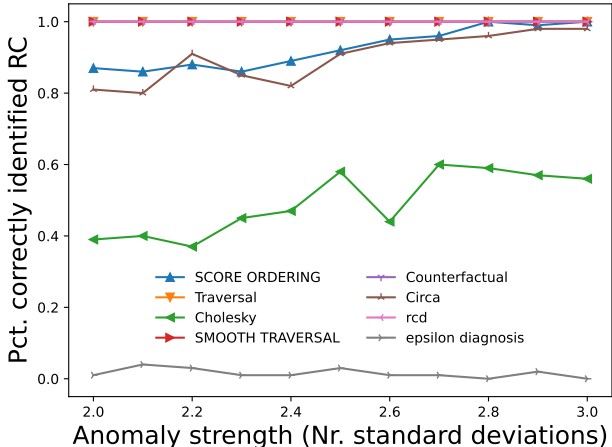

Figure 6: True positive rate for identifying the root cause against anomaly strength injected at the root cause, when all causal graphs are polytrees. Note that RCD and $\varepsilon$-Diagnosis are given 100 samples from the anomalous period, while all other algorithms are given only one.

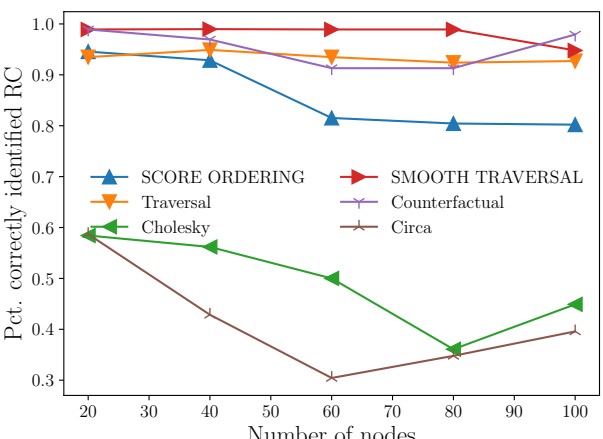

Figure 7: True positive rate of the compared algorithms in identifying the root cause with increasing numbers of nodes.

strength of $x = 3.0$, all structural equations linear, and the graph provided to each algorithm randomly altered to introduce mismatches with the true causal graph.

Starting from the sampled, true causal graph, random edge additions, removals and reversals were introduced according to a desired target Structural Hamming Distance [57, 58] from the true graph. To control for the density of the graph, an equal number of edges are randomly added as removed. In particular, for true causal graph $\mathcal{G} = (V, E)$, the total number of edges added and removed is sampled uniformly at random from $n_{\mathrm{add/rem}} = [\mathrm{SHD}(\mathcal{G}, \mathcal{G}_{alt}) - |E|, \ \mathrm{SHD}(\mathcal{G}, \mathcal{G}_{alt})]$, with the number of edges flipped $n_{\mathrm{flip}} = \mathrm{SHD} - 2 \cdot n_{\mathrm{add/rem}}$. Of the edges in $\mathcal{G}$, $n_{\mathrm{add/rem}}$ are randomly selected for removal, and an equal number of 'missing' edges are selected for addition. Of the remaining 'un-removed' edges in $\mathcal{G}$, $n_{\mathrm{flip}}$ are reversed. If the resulting graph is not a DAG, the whole process is repeated until it is. The procedure for generating a randomly misspecified graph given a ground truth matches the "edge domain expert" procedure described in [59].

We chose to sample all linear structural equations as the relationship between top-1 recall and SHD was the same as when sampling non-linear structural equations for SMOOTH TRAVERSAL, Traversal and Counterfactual, but was more noteworthy for Circa. The reason being that the performance for

Circa is already so low for non-linear structural equations, that a considerable drop in performance was not easily perceptible.

We observe that all four evaluated algorithms show a considerable drop in performance as the SHD of the given graph from the true causal graph increased, with the most pronounced drop occurring for Circa. However, even when half of all edges are removed/flipped or added elsewhere in the graph (SHD/$|E| = 1$), SMOOTH TRAVERSAL, Traversal and Counterfactual still achieve a top-1 recall of approximately 0.6 to 0.7, showing that all three algorithms are relatively robust to graph misspecification.

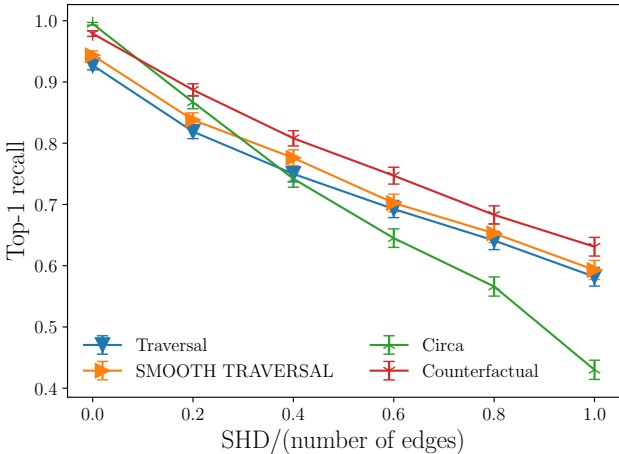

Figure 8: Top-1 recall for identifying the root cause using a graph with an increasing SHD (average per edge) from the true causal graph.

### F.5    How do the algorithms compare in terms of their practicality?

It is important to keep in mind when interpreting the performances of each of the algorithms their different input requirements. Counterfactual either needs to be provided the SCM or must attempt to learn it from data. Traversal, Circa and SMOOTH TRAVERSAL require the causal graph as input but with differing assumptions on its structure. Traversal makes no assumptions, Circa assumes the SCM is linear and attempts to learn it from data, given the graph, and SMOOTH TRAVERSAL has guarantees only for the case the graph is a polytree (though it may be applied even if this assumption is not met). Traversal, however, requires an additional tuning parameter: the anomaly score threshold, while SMOOTH TRAVERSAL does not have any such free parameter. Both SCORE ORDERING and Cholesky do not even require the causal graph, however, Cholesky assumes the underlying SCM is linear. In synthetic data experiments, SCORE ORDERING consistently outperformed Cholesky other than when the SCM was restricted to be linear (see Fig. 5 above). Cholesky also scales more poorly in terms of run-time (see Fig. 3) due to the need to estimate the covariance matrix. In comparison, SCORE ORDERING makes very weak demands on the input, is efficient, and consistently performs well across experiments.

### F.6    PetShop dataset

Hardt et al. [27] introduce a dataset from cloud computing, specifically designed for evaluating root cause analyses in microservice-based applications. The dataset is collected from a microservice application and includes 68 injected performance issues, which increase latency and reduce availability throughout the system. Latency and availability are so-called 'metrics' of the microservice application, and measure the length of time it takes for a request to the service to be processed, and the fraction of the time a request returns an error, respectively. The dataset also encompasses three traffic scenarios: low, high, and temporal. These correspond to the amount and periodicity of user traffic to the application. This presents a challenging task due to high missingness, low sample sizes, and near-constant variables. Furthermore, the ground truth causal graph is only partially known, as we must treat the edge-inverted call graph as a proxy for the causal graph.

In addition to the approaches evaluated by Hardt et al. (see [27] for more details), which we have reproduced below, we evaluated our algorithms in both top-1 recall (Table. 1) and top-3 recall (Table. 2). In addition, we included, as an additional baseline, the Traversal algorithm as described in Section 4. This is added as although the evaluation already includes a 'Traversal' algorithm implemented by Hardt et al., here called 'petshop Traversal', their algorithm does not match the standard approach as described in [9, 24, 25]. Instead, 'petshop Traversal' has been designed for use on the PetShop dataset, takes into account particular aspects of that data, and contrary to standard traversal algorithms, scores potential root causes according to anomalousness of whole paths of nodes from root cause to target node. 'Petshop Traversal' should not, therefore, be treated as an 'out-of-the-box' baseline.

We observe that SMOOTH TRAVERSAL and SCORE ORDERING perform very well in top-1 and top-3 recall, often being the best performing algorithms, and clearly outperforming simple Traversal and Cholesky. However, while SMOOTH TRAVERSAL always matches or exceeds the performance of simple Traversal in top-1 recall, on latency issues it nonetheless tends to perform relatively poorly compared to SCORE ORDERING.

Table 1: Top-1$^*$ recall, with ties. $^*$Multiple anomalies can receive the same, highest, anomaly score. The root cause being in the top-1 means that it is among the variables receiving the highest score.

| traffic scenario | metric | graph given | | | graph not given | | | | | this paper | |
|---|---|---|---|---|---|---|---|---|---|---|---|
| | | petshop Traversal | Circa | counter-factual | $\epsilon$-diagnosis | rcd | correlation | Traversal | Cholesky | SCORE ORDERING | SMOOTH TRAVERSAL |
| low | latency | 0.57 | 0.36 | 0.36 | 0.00 | 0.07 | 0.43 | 0.14 | 0.29 | **1.00** | 0.14 |
| low | availability | 0.50 | 0.42 | 0.00 | 0.00 | 0.58 | **0.75** | 0.42 | 0.00 | **0.75** | **0.75** |
| high | latency | 0.57 | 0.50 | 0.57 | 0.00 | 0.00 | 0.64 | 0.14 | 0.14 | **1.00** | 0.14 |
| high | availability | 0.33 | 0.00 | 0.00 | 0.00 | 0.00 | 0.83 | 0.33 | 0.00 | **1.00** | **1.00** |
| temporal | latency | **1.00** | 0.75 | 0.38 | 0.12 | 0.25 | 0.62 | 0.25 | 0.50 | 0.75 | 0.25 |
| temporal | availability | 0.38 | 0.38 | 0.00 | 0.00 | 0.50 | 0.62 | 0.25 | 0.00 | **1.00** | **1.00** |

Table 2: Top-3$^*$ recall, with ties. $^*$Multiple anomalies can receive the same, highest, anomaly score. The root cause being in the top-3 means that it is among the variables ranked in the top three including all variables tied at the top.

| traffic scenario | metric | graph given | | | graph not given | | | | | this paper | |
|---|---|---|---|---|---|---|---|---|---|---|---|
| | | petshop Traversal | Circa | counter-factual | $\epsilon$-diagnosis | rcd | correlation | Traversal | Cholesky | SCORE ORDERING | SMOOTH TRAVERSAL |
| low | latency | 0.57 | 0.86 | 0.71 | 0.00 | 0.21 | 0.57 | 0.14 | 0.57 | **1.00** | 0.86 |
| low | availability | **1.00** | **1.00** | 0.42 | 0.00 | 0.75 | 0.92 | 0.50 | 0.00 | **1.00** | **1.00** |
| high | latency | 0.79 | **1.00** | 0.86 | 0.00 | 0.07 | 0.79 | 0.14 | 0.57 | **1.00** | 0.93 |
| high | availability | **1.00** | 0.00 | 0.00 | 0.33 | 0.00 | 0.92 | 0.33 | 0.00 | **1.00** | **1.00** |
| temporal | latency | **1.00** | **1.00** | 0.50 | 0.12 | 0.75 | 0.75 | 0.25 | 0.63 | **1.00** | 0.75 |
| temporal | availability | **1.00** | **1.00** | 0.25 | 1.00 | 0.12 | 0.75 | 0.25 | 0.00 | **1.00** | **1.00** |

Note, however, the caveat that owing to the small sample size in the normal period in the PetShop dataset, it is frequently the case that multiple anomalies receive the same highest IT anomaly score (see Table 3 for numbers and proportions of nodes with tied scores in eah traffic scenario). In the most severe cases there can be as many as 20 nodes tied with the maximum score (in the high traffic scenario), but it is typically fewer than 10. The multiplicity of highest scores is due to the fact that for the issue at hand all these metrics were more extreme than ever sampled in the normal period. In this case, the root cause being in the top-1 means that it is among the variables receiving the highest score. This problem should become less severe for datasets where anomaly scorers are trained on a longer history. Since root cause analysis without any graphical information is a notoriously hard problem, returning short lists of potential root causes, even if they contain 10 or more candidates, can still result in a drastic reduction of the search space, and so is of vital use in real applications.

Table 3: Average number and proportion of nodes in each traffic scenario and metric with a tied marginal anomaly score

| Traffic scenario | Metric | Average number | Average proportion |
|---|---|---|---|
| high traffic | Latency | 18.26 | 41.50% |
| high traffic | Availability | 16.46 | 37.41% |
| low traffic | Latency | 9.96 | 23.72% |
| low traffic | Availability | 6.54 | 15.57% |
| temporal traffic1 | Latency | 5.75 | 14.02% |
| temporal traffic1 | Availability | 7.00 | 17.07% |
| temporal traffic2 | Latency | 5.75 | 14.02% |
| temporal traffic2 | Availability | 7.12 | 17.38% |

We, nonetheless, also report the top-1 (Table. 4) and top-3 (Table. 5) recall after randomly selecting among the nodes tied with the highest anomaly score. As expected, we observe an approx. 10x decrease in top-1 recall for Traversal, SCORE ORDERING and SMOOTH TRAVERSAL, corresponding to selecting at random from among the approx. 10 candidate root causes with tied highest scores. Similar decreases are present in top-3 recalls for Traversal and SCORE ORDERING whenever the true root cause is among those with the tied highest score. Note, however, that SMOOTH TRAVERSAL generally still has high top-3 recall even after random selection of the top-1 node, outperforming both Traversal, SCORE ORDERING and Cholesky in all but one evaluation. Note that the performance of Cholesky does not considerably drop after random selection of ties, as variables rarely receive the same score.

Table 4: Top-1 recall, with random selection among ties.

| traffic scenario | metric | Traversal | Cholesky | this paper | |
|---|---|---|---|---|---|
| | | | | SCORE ORDERING | SMOOTH TRAVERSAL |
| low | latency | 0.02 | **0.29** | 0.10 | 0.02 |
| low | availability | 0.06 | 0.00 | **0.10** | **0.10** |
| high | latency | 0.01 | **0.14** | 0.06 | 0.01 |
| high | availability | 0.03 | 0.00 | 0.06 | **0.09** |
| temporal | latency | 0.04 | **0.50** | 0.10 | 0.04 |
| temporal | availability | 0.04 | 0.00 | 0.13 | **0.14** |

Table 5: Top-3 recall, with random selection among ties

| traffic scenario | metric | Traversal | Cholesky | this paper | |
|---|---|---|---|---|---|
| | | | | SCORE ORDERING | SMOOTH TRAVERSAL |
| low | latency | 0.02 | 0.57 | 0.10 | **0.73** |
| low | availability | 0.10 | 0.00 | **0.31** | **0.31** |
| high | latency | 0.01 | 0.50 | 0.06 | **0.80** |
| high | availability | 0.03 | 0.00 | 0.06 | **0.09** |
| temporal | latency | 0.04 | **0.63** | 0.18 | 0.54 |
| temporal | availability | 0.04 | 0.00 | 0.13 | **0.14** |

### F.7 Sock-shop dataset

Similar to the PetShop dataset described above, Pham et al. [26] introduces another dataset from cloud computing for evaluating root cause analyses. The dataset includes 125 injected performance issues encompassing five types of common faults: CPU hog - 'cpu', memory leak - 'mem', disk IO stress - 'disk', network delay - 'delay', and packet loss - 'loss'. We again used the edge-inverted call graph for the Sock-shop application as a proxy for the causal graph. We evaluated the performance

of our algorithms in top-1 (Table 6) and top-3 (Table 7) recall. It is important to note that while for most algorithms we provided only a single anomalous sample (the first sample from the anomalous period) for RCD and $\epsilon$-Diagnosis we provided all 361 anomaly samples and so direct comparison is not recommended: both algorithms were included rather for completeness of evaluation.

We observe that on three out of five issue types (delay, disk and loss), SCORE ORDERING is the top performing approach for top-1 and top-3 recall. In each instance, SMOOTH TRAVERSAL also performed competitively. For issue types cpu and mem, however, both SCORE ORDERING and SMOOTH TRAVERSAL performed poorly, although performance was comparable among all algorithms which were provided with only a single anomalous sample. Given both RCD and $\epsilon$-Diagnosis performed considerably better for both issues, this would suggest that the effect of the fault was not easily discernible from only a single sample in the anomaly period.

As for PetShop, we also report the top-1 (Table 8) and top-3 (Table 9) recall after randomly selecting among nodes with tied ranks. Here we observe that although the performance of SMOOTH TRAVERSAL is competitive for delay, disk and loss issues, both SCORE ORDERING and SMOOTH TRAVERSAL are outperformed by Circa and/or Cholesky. As both algorithms assume linear causal models, and that both algorithms struggled in non-linear settings in our simulation studies (see Figure 1), this may suggest that the causal relationships in Sock-shop are well modelled by linear relationships. As before, for cpu and mem issues, while all methods which use only a single anomaly sample perform very poorly, RCD performs well.

Table 6: Top-1$^*$ recall, with ties. $^*$Multiple anomalies can receive the same, highest, anomaly score. The root cause being in the top-1 means that it is among the variables receiving the highest score. $^\dagger$These methods were provided with all samples from the anomaly period, whereas all else were given only one.

|                           | cpu      | delay    | disk     | loss     | mem      |
| ------------------------- | -------- | -------- | -------- | -------- | -------- |
| SCORE ORDERING            | 0.08     | **0.80** | **0.84** | **0.72** | 0.08     |
| SMOOTH TRAVERSAL          | 0.00     | 0.72     | 0.76     | 0.60     | 0.00     |
| Cholesky                  | 0.04     | 0.56     | 0.48     | 0.40     | 0.12     |
| Circa                     | 0.04     | 0.52     | 0.68     | 0.64     | 0.16     |
| Counterfactual            | 0.04     | 0.28     | 0.00     | 0.12     | 0.04     |
| Traversal                 | 0.04     | **0.80** | 0.76     | 0.72     | 0.12     |
| rcd$^\dagger$             | **0.88** | 0.52     | 0.52     | 0.44     | **0.36** |
| epsilon diagnosis$^\dagger$ | 0.24   | 0.08     | 0.16     | 0.20     | 0.00     |

Table 7: Top-3$^*$ recall, with ties. $^*$Multiple anomalies can receive the same, highest, anomaly score. The root cause being in the top-3 means that it is among the variables ranked in the top three, including tied ranks. $^\dagger$These methods were provided with all samples from the anomaly period, whereas all else were given only one.

|                           | cpu      | delay    | disk     | loss     | mem      |
| ------------------------- | -------- | -------- | -------- | -------- | -------- |
| SCORE ORDERING            | 0.24     | **0.92** | **0.92** | **0.88** | 0.16     |
| SMOOTH TRAVERSAL          | 0.12     | 0.80     | 0.84     | 0.72     | 0.12     |
| Cholesky                  | 0.16     | 0.88     | 0.88     | 0.84     | 0.24     |
| Circa                     | 0.28     | 0.72     | 0.80     | 0.80     | 0.28     |
| Counterfactual            | 0.24     | 0.76     | 0.52     | 0.64     | 0.40     |
| Traversal                 | 0.04     | 0.80     | 0.76     | 0.72     | 0.12     |
| rcd$^\dagger$             | **1.00** | 0.52     | 0.56     | 0.44     | **0.52** |
| epsilon diagnosis$^\dagger$ | 0.56   | 0.52     | 0.28     | 0.48     | 0.00     |

Table 8: Top-1 recall, with random selection among ties. †These methods were provided with all samples from the anomaly period, whereas all else were given only one.

|  | cpu | delay | disk | loss | mem |
|---|---|---|---|---|---|
| SCORE ORDERING | 0.08 | 0.24 | 0.29 | 0.26 | 0.04 |
| SMOOTH TRAVERSAL | 0.00 | 0.49 | 0.50 | 0.47 | 0.00 |
| Cholesky | 0.04 | **0.56** | 0.48 | 0.40 | 0.12 |
| Circa | 0.04 | 0.52 | **0.68** | **0.64** | 0.16 |
| Counterfactual | 0.04 | 0.28 | 0.00 | 0.12 | 0.00 |
| Traversal | 0.02 | 0.41 | 0.37 | 0.47 | 0.02 |
| rcd† | **0.88** | **0.56** | 0.48 | 0.40 | **0.36** |
| epsilon diagnosis† | 0.24 | 0.08 | 0.16 | 0.20 | 0.00 |

Table 9: Top-3 recall, with random selection among ties. †These methods were provided with all samples from the anomaly period, whereas all else were given only one.

|  | cpu | delay | disk | loss | mem |
|---|---|---|---|---|---|
| SCORE ORDERING | 0.24 | 0.67 | 0.70 | 0.68 | 0.14 |
| SMOOTH TRAVERSAL | 0.12 | 0.76 | 0.83 | 0.72 | 0.04 |
| Cholesky | 0.16 | **0.88** | **0.92** | **0.84** | 0.24 |
| Circa | 0.28 | 0.72 | 0.80 | 0.80 | 0.28 |
| Counterfactual | 0.24 | 0.80 | 0.48 | 0.64 | 0.28 |
| Traversal | 0.04 | 0.79 | 0.75 | 0.72 | 0.07 |
| rcd† | **1.00** | 0.56 | 0.56 | 0.44 | **0.48** |
| epsilon diagnosis† | 0.56 | 0.52 | 0.28 | 0.48 | 0.00 |

## F.8 ProRCA

To provide evaluations of the algorithms in a semi-synthetic setting, we used the ProRCA package, introduced in [48]. ProRCA consists of a hand-crafted model of a retail service, with a known causal graph, allowing synthesis of data from real-world business scenarios, into which different types of anomalies can be introduced.

There are five types of anomalies that can be injected in the process, affecting different variables. We synthesised data following the introduction of four of these anomaly types, as listed in the first column in Table 10, performing 100 replicates of each. We excluded the 'COGs' anomaly type, which introduces an anomaly at the variable 'UNIT_COST', as for a range of strengths of the anomaly, we were unable to produce changes in the data that were statistically distinct from those in the non-anomalous regime.

We removed from consideration variables in the graph that had plain text values, such as 'PROMO_CODE' as these are unsuitable for RCA, but doing so introduced no unmeasured confounding and so does not affect the analysis. For each anomaly, we considered 'PROFIT' to be the target variable. 'PROFIT' is not a leaf node in the causal graph and the length of the paths between the root cause and the target variable varies between two and three. We applied Circa, Counterfactual, Traversal, and SMOOTH TRAVERSAL algorithms for which we are able to provide (the marginalised) causal graph, and the Cholesky algorithm and SCORE ORDERING, which only requires data from the observational and the anomalous regime. Only one anomalous sample was provided in each case. The resulting Top-1 recall of the algorithms is reported in Table 10.

The results are mixed. SCORE ORDERING performs well for two types of anomaly. In both cases, the methods that require the graph also perform well. However, requiring the graph for the RCA algorithm might be constraining in most real-world scenarios. There are two types of anomaly where SCORE ORDERING does not perform well: In 'ExcessiveDiscount' only the Counterfactual method provides a Top-1 recall above 30%, even for those methods requiring the graph as input. In 'ReturnSurge', the Cholesky method performs well (66%) and those methods for which the graph

Table 10: Top-1 recall, with random selection among ties.

| Type of anomaly | graph given | | | graph not given | this paper | |
| | Circa | Counterfactual | Traversal | Cholesky | SCORE ORDERING | SMOOTH TRAVERSAL |
|---|---|---|---|---|---|---|
| ExcessiveDiscount | 0.04 | **0.89** | 0.30 | 0.00 | 0.04 | 0.04 |
| FulfillmentSpike | 0.96 | **0.98** | 0.94 | 0.74 | 0.94 | 0.28 |
| ReturnSurge | **1.00** | 0.88 | 0.97 | 0.66 | 0.00 | 0.18 |
| ShippingDisruption | 0.94 | 0.94 | 0.94 | 0.47 | **0.95** | 0.27 |

is given provide a recall above 85%. In all types of anomalies, SMOOTH TRAVERSAL has a performance below 30%.

