# OpenReview forum: "Root Cause Analysis of Outliers with Missing Structural Knowledge"
_NeurIPS.cc/2025/Conference — NeurIPS 2025 poster_

### Official Review · Reviewer_juNp · 2025-06-12

**Clarity:** 2
**Significance:** 3
**Originality:** 2
**Rating:** 3
**Confidence:** 3

**Summary:**

Root Cause Analysis of Outliers with Missing Structural Knowledge tackles the challenge of identifying root causes from a single anomaly when the full causal graph is unknown. The authors define each variable’s ‘surprise’ as the negative log tail-probability of a chosen statistic, forming an information-theoretic (IT) anomaly score derived solely from background data. In directed-tree (polytree) structures, sharp increases in these scores indicate the origin of change. Building on this, Smooth Traversal traces back from the anomalous node to its highest-scoring ancestor using a known DAG. When the graph is unknown, they introduce Score Ordering, which ensures the true root ranks among the top-k scorers under a mild in-degree assumption. Experiments on synthetic and real-world data show these efficient methods outperform stronger baselines with significantly less structural knowledge and computation.

**Questions:**

1) If the authors are constrained by the page limit, it may be helpful to adjust the formatting to save space and allow for a more comprehensive conclusion, clearer discussion, and better-organized experimental section?
2) Would it be possible to include more comprehensive experiments using additional real-world datasets, as well as scenarios where the proposed methods might fail?
3) Would it be possible to expand the conclusion and reorganize the introduction to improve clarity and readability?

**Ethical Concerns:**

["NO or VERY MINOR ethics concerns only"]

**Final Justification:**

The authors have addressed my concerns; however, the novelty and technical quality still do not warrant a higher score, so I am maintaining my original rating.

**Limitations:**

No. While the paper clearly names several technical caveats—e.g., dependence on a polytree structure, the single-fault setting, and the need for sufficient background data (Sec. 3–4)—it does not meaningfully discuss (i) how these assumptions limit real-world deployment, or (ii) Sample-complexity guidance.
1) While the authors acknowledge the limitations of these strong assumptions, they do not thoroughly discuss their implications in practical applications and real-world scenarios.
2) Sample-complexity guidance. Provide empirical or theoretical bounds on the amount of background data required for stable information-theoretic scores, so practitioners can gauge feasibility before adoption.

**Paper Formatting Concerns:**

NAN

**Quality:**

2

**Strengths And Weaknesses:**

Strengths
1) Timely problem setting with clear practical impact – the paper tackles RCA when only one anomalous sample is available and conditional densities are ill-posed to learn, a scenario common in fast-moving operations and personalised medicine .
2) Conceptual originality. They re-casting outlier “surprise” as an information-theoretic (IT) anomaly score that needs only background data removes the hardest estimation bottleneck.
3) Algorithmic simplicity & efficiency – SMOOTH TRAVERSAL and SCORE ORDERING run in near-linear time and need either the DAG or just an in-degree bound; experiments show they are the fastest among tested methods while matching or exceeding stronger baselines on accuracy .

Weaknesses
1) The overall presentation is difficult to follow without close reading. For example, in the experimental section, it is not easy to clearly identify the model, dataset, and settings used, or what the author aims to demonstrate. It would be beneficial to organize these components with clearer logic and structure. It appears that the author may be constrained by the page limit—yet there are formatting inefficiencies, such as on lines 243, 213, and 67, where single words occupy entire lines. These could be optimized to save space. Additionally, the Introduction and Conclusion are very brief but not in a concise or effective way that helps readers quickly grasp the paper's core contributions and content.
2) While the authors repeatedly state in the methodology section that the 'Monotonicity Assumption' is not required, they continue to rely on it for demonstrations. It would be beneficial, if possible, to present general results in main paper without this assumption, as it is often unrealistic in practical settings.
3) The experiment part can be expand if more real word dataset can be evaluated to show the robustness of the methods proposed by author. And if possible, further analysis on case study when will the methods fails to guide practical application since this is one of the main motivation brought by authors.

---

> ### Author Rebuttal · Authors · 2025-07-31
>
> > Addressing weaknesses
>
> 1. Thank you for raising this issue, it’s of course very important that the paper is as easy to follow as possible. For the Experiments section we propose stating, in bullet points, for each experiment: how it is performed, why it is performed, the range of synthetic parameters used and the methods compared including the assumptions each makes. In addition, we propose including a new table which summarises this information and where to find the corresponding results figures. Thank you as well for pointing out where we can optimise space. For the Introduction, we propose reorganising Subsection 1.1 to give a more detailed breakdown of the following sections, including a summary of the main results from each, and the motivations for each. We hope this will better motivate the work and allow the reader to better anticipate what results are coming and why. Finally, you are absolutely right that the Conclusion is unhelpfully brief. This was an unfortunate mistake arising from last minute changes to meet the page limit. We will expand the conclusion, and suggest the following: “We have explored several directions in which the practical limitations of RCA can be addressed: first by avoiding estimation of conditional distributions in the setting that the causal graph is known, and second by avoiding needing the causal graph at all, in the setting that it is a polytree. To do so, we leverage information theoretic (IT) anomaly scores, and prove general laws about their typical decay along causal paths. Using our results, we then propose two novel algorithms: SMOOTH TRAVERSAL and SCORE ORDERING, give probabilistic guarantees for the correctness of their outputs, and show they have competitive performance in both synthetic and real-world data despite their simplicity. To our knowledge, our work is the only which provides guarantees for RCA in the non-linear setting from a single anomalous point without requiring knowing the full structural causal model.”
>
> 2. You are absolutely right that monotonicity is unrealistic in practical settings, and you have noticed how important it was to us to stress that this assumption is not required, hence the repeated statements throughout Section 3. We felt it was very unfortunate that we had to reserve the theory without the assumption for Appendix B. This was done because we worried that the theory would become more difficult to follow as all of the Lemmas and Theorems following the assumption must be stated more intricately, and the bounds presented become slightly less concise as the inequalities hold up to some small $\epsilon$. We accept your criticism nonetheless. We propose instead to move Lemma B.2 from Appendix B into the main text so as to explain why the inequalities hold up to some $\epsilon$, and subsequently state the results as they are currently presented but without monotonicity and using the slightly less concise inequalities. We will then continue to refer to the Appendix for the full detail for each Lemma and Theorem without monotonicity. This should be possible with the extra space afforded by the camera-ready version of the manuscript, should it be accepted, without significant loss of clarity.
>
> 3. You are right to raise the issue of robustness. To expand our evaluation to more real-world scenarios, we will in addition to the experiments we have already included using the PetShop dataset and the ProRCA package, we will evaluate each of the algorithms on the Sock-shop 2 dataset (Pham et al, 2024, Root Cause Analysis for Microservice System based on Causal Inference: How Far Are We?). We will also expand the synthetic experiments in two ways: first we will repeat the experiments, but restricting the causal graph to be polytree or a collider-free polytree (as currently in no experiment to impose this constraint) to see how performance may be affected when this assumption is met, second, we propose comparing the performance of each of the algorithms which depend on being given the causal graph (i.e. Traversal, Counterfactual, SMOOTH TRAVERSAL, and Circa) after introducing an increasing number of random edge additions/deletions/reversals to the graph they are given, simulating the case where the assumed graph does not perfectly match the true graph. This will not affect the performances of SCORE ORDERING and Cholesky which do not rely on the causal graph, but should allow us to test how robust SMOOTH TRAVERSAL is to misspecification of the graph compared to the other algorithms, and to assess at what point it is preferable to use a method which does not rely on the causal graph, such as SCORE ORDERING, compared to one that does. Finally, we will include an expanded discussion on when we expect our proposed algorithms to fail with a focus on guiding practical application. In particular, we will discuss what our theory shows about the difficulty of RCA in settings where the causal graph is very large (as note that both the bounds in Theorem 3.9 and Theorem 3.10 become weaker for larger $n$, though note as well that this challenge is not unique to our algorithms), and when the sample size for the normal period is low. Note that in Subsection 3.5 and Appendix D we discuss one particularly important case where the polytree assumption is violated and there is strong confounding, and why in practice it may not pose a significant problem. We will move some of the discussion from Appendix D into the main text as there we show that we expect SCORE ORDERING to fail only rarely when the polytree assumption is violated (at least in the linear setting).
>
> > Addressing questions
>
> 1. Please see our reply to point 1 in "Addressing weaknesses" above
> 2. Please see our reply to point 3 in "Addressing weaknesses" above
> 3. Please see our reply to point 1 "Addressing weaknesses" above
>
> > Addressing limitations
>
> 1. This is a very important point, I hope that you will feel that our reply to point 3 in "Addressing weaknesses" above already largely  encompasses the issue you raise here. Nonetheless, let us provide some further information on two points. First, the polytree assumption: the polytree assumption is indeed restrictive ). In subsection 3.5 and Appendix D we give some justification for the application of our algorithms even in cases where this assumption is violated, and in all experiments we do not restrict the causal graph to be a polytree, and both of our algorithms show competitive performance. Note also that the only other Root Cause Analysis approach which operates in the setting we study (Li et al, 2025, “Root cause discovery via permutations and Cholesky decomposition”), the method we refer to as “Cholesky” in the Experiments section, assumes that the causal model is linear, while our approach does not. As such, we still believe that our work is an important contribution to the current state of root cause analysis research, despite this assumption. Regarding the single fault setting: if this is referring to our assumption that there is only a single root cause, in fact our theory readily generalises to the case where there are multiple. Olease see the detailed reply we gave to reviewer D5Fu above concerning this (and sorry for sending you to one of our replies to another reviewer, they raised the same point, and we are running out of characters!).
>
> 2. Thank you for raising the point of sample-complexity guidance. Given as low sample sizes in the normal period can pose issues for use of estimated IT anomaly scores in practice, it would certainly be worth providing some guidance on this. We propose including the following additional result in the appendix, with proof: if we use at least $k =\frac{3 e^{S_{\max}}}{\delta^2} \log\left(\frac{2n}{\alpha}\right)$ samples, then the estimated anomaly scores $\hat{S}(x_i)$ satisfy $P\left(|\hat{S}(x_i) - S(x_i)| < \delta \right) \quad \forall i \in \{1, \dots, n\}$ with probability at least $1 - \alpha$, where $S_{\max} := \max_i S(x_i)$, and $S(x_i)$ and $\hat{S}(x_i)$ are as defined in Equations 4 and 5 in the main text. The result follows from applying the (multiplicative form of) the Chernoff bound for small $\delta$, the full proof of which we will provide in the appendix.

---

> > ### Comment · Reviewer_juNp · 2025-08-05
> > **Response to Rebuttal**
> >
> > **Causal Graph Perturbation Validity**
> > 1. The proposed robustness evaluation with graph perturbations is very promising. To ensure interpretability, could the authors clarify whether the perturbed graphs remain valid DAGs? For example, do the edge reversals or additions guarantee acyclicity? This detail would help strengthen the conclusions drawn from this part of the study.
> >
> > **Application to Black-box Systems**
> > 2. In many real-world scenarios, the causal graph is not readily available and must be estimated. I wonder if the authors envision integrating a causal discovery step into their framework. If so, it would be helpful to briefly discuss how that might affect the theoretical guarantees or empirical robustness of the proposed methods.

---

> > > ### Author Response · Authors · 2025-08-09
> > > **Preliminary graph perturbation results**
> > >
> > > We ran a preliminary analysis for the graph perturbation experiments described in the previous comment above and present the results in a table below. We use the same synthetic data generator as in the paper for Figure 1 (anomaly strength = 3.0, number of nodes = 30) and display the average top-1 recall, over 50 replicates, after introducing random edge changes (quantified by the SHD/num_edges in the columns).
> > >
> > > | method \ SHD/num_edges | 0.0 | 0.5 | 1.0 | 1.5 | 2.0 |
> > > | --- | --- | --- | --- | --- | --- |
> > > | SMOOTH TRAVERSAL | 0.960 | 0.960 | 0.860 | 0.860 | 0.740 |
> > > | Counterfactual | 0.920 | 0.940 | 0.860 | 0.960 | 0.760 |
> > > | Traversal | 0.880 | 0.820 | 0.780 | 0.780 | 0.600 |
> > > | Circa | 0.560 | 0.500 | 0.440 | 0.540 | 0.720 |
> > >
> > > We see that all of the evaluated methods are relatively robust to misspecification of the causal graph, with Traversal suffering the most, and Circa actually improving with increasing SHD. Comparing to Figure 1 in the paper, this shows that for moderate misspecification of the graph, it may still preferable to use SMOOTH TRAVERSAL over SCORE ORDERING, but for stronger perturbations the situation flips. We would perform a more thorough analysis, with more replicates, in the revised manuscript to see if these conclusions hold over increasing graph sizes and anomaly strengths.

---

> ### Author Response · Authors · 2025-08-06
>
> **Causal Graph Perturbation Validity**
>
> That is a good point, yes that would absolutely be the case. We propose the following outline approach to graph perturbations: generate a random “ground truth” DAG, introduce random edge additions/deletions/reversals subject to a given Structural Hamming Distance (SHD) constraint, check if acyclic (and if not, reject and repeat), evaluate each algorithm using the perturbed graph. The idea is to enforce acyclicity via rejection sampling, and to enforce how “big” the perturbation is by fixing the SHD from the ground truth graph (in practice this is simple as it need only count the number of random edge changes we introduce). This can then be repeated many times for several SHD values. This way we can precisely test how robust each of the methods are to increasingly large perturbations from ground truth. The SHD is convenient for quantifying perturbation size as it is the standard metric in the literature for evaluating the performance of causal discovery algorithms, and so our results should be directly comparable to those in the literature if one is choosing a causal discovery algorithm.
>
> **Application to Black-box Systems**
>
> This is a good question. You are right that causal graphs are often unavailable in practice, so a natural suggestion would be to perform causal discovery first. The first thing to note is that SMOOTH TRAVERSAL and SCORE ORDERING are closely related and follow from the same theoretical insights into anomaly propagation (Lemmas 3.2, 3.7, B.1 and B.2). In this sense, SCORE ORDERING is akin to a graph-free “version” of SMOOTH TRAVERSAL. So rather than introducing a potentially unreliable causal discovery step, our recommendation would be to use SCORE ORDERING when the causal graph is unknown.
>
> That said, as your question on robustness importantly highlights, the situation is rarely binary. One may have a causal graph (e.g. from domain expertise) but not know whether it can be trusted, and in which case, not know which of our proposed algorithms should be used. The theory we present suggests a way to assess it: anomalies scores should propagate along edges and generally decay along causal paths, so if one observes many large score “jumps” from parent to child, or disconnected anomalies, this would suggest there are orientation errors in the given graph unless the “sparse mechanism shift hypothesis” (Schölkopf et al. 2021) is violated. SCORE ORDERING may then be preferable. We will certainly include a discussion of this point in the revised manuscript.
>
> With this in mind, we don’t propose integrating a causal discovery step into our framework for three main reasons: one because of the above, two as we worry that doing so would limit its application by committing to a particular existing causal discovery algorithm, and three, relatedly to points one and two, existing causal discovery algorithms tend to be highly unreliable. Finally, while formal robustness guarantees would likely be challenging to obtain (we are not aware of any such results in existing RCA literature) we hope the newly proposed experiments will thoroughly address this empirically!

---

### Official Review · Reviewer_RvMn · 2025-06-15

**Clarity:** 3
**Significance:** 3
**Originality:** 2
**Rating:** 4
**Confidence:** 3

**Summary:**

The paper addresses the problem of RCA in scenarios where only a single sample from the anomalous distribution is available, relying on information-theoretic anomaly scores. The author acknowledges that, in practice, anomalies often have multiple root causes; however, they argue that these can be addressed sequentially, one at a time. Two methods are proposed: (1) Smooth Traversal – Applicable when the causal graph is known and structured as a polytree. This method identifies the root cause by selecting the variable that shows the largest increase in anomaly score relative to its parents. (2) Score Ordering – Designed for settings where the causal graph is unknown. It assumes that the top-k variables with the highest marginal anomaly scores are likely to be root causes. The paper provides theoretical guarantees for both methods under specific assumptions. Experiments on both synthetic and real-world datasets demonstrate the effectiveness of the proposed.

**Questions:**

1. From the results of the experiments, baselines Traversal and Counterfactual have comparable performance with Smooth Traversal, maybe more complex dataset is needed.
2. Please add more detail about the difference of this work from the following work: ‘[23] Kailash Budhathoki, Lenon Minorics, Patrick Bloebaum, and Dominik Janzing. Causal structure based root cause analysis of outliers’
3. Could the authors clarify how Equation (5) is computed or estimated in practice?
4. In real-world scenarios, it is common to observe multiple anomalous samples rather than just a single one, as this helps reduce false alarms. Given this, how should the proposed method be applied when multiple anomalous samples are available? Since different samples might lead to different conclusions,

**Ethical Concerns:**

["NO or VERY MINOR ethics concerns only"]

**Final Justification:**

The problem setting of identifying the root cause from a single anomalous sample is interesting. In the rebuttal, the authors shared some promising ideas for extending the approach to handle multiple anomalies; however, I still find the underlying assumption somewhat restrictive. Overall, I will maintain my score.

**Limitations:**

The authors have addressed key limitations of their method. They also suggest that their methods could serve as heuristics in broader contexts. However, the author could include a brief discussion on potential societal consequences, especially in high-stakes fields like healthcare or finance, where erroneous root cause identification could lead to harm.

**Quality:**

3

**Strengths And Weaknesses:**

Strengths:
(1) It addresses a practical challenge by focusing on RCA with a single anomalous sample, a scenario very close to real-world settings.
(2) The methods are computationally simple and efficient.
(3) The structure of the paper is clear and logical, progressing smoothly from background material to theoretical results and empirical validation.


Weaknesses:
(1) The assumption of polytree is a kind of restrictive, as real-world causal structures are often more complex (e.g., containing cycles or multiple parents).
(2) Experimental results show that baseline methods such as Traversal and Counterfactual perform comparably to Smooth Traversal (authors’), suggesting that more complex or varied datasets may be needed to demonstrate clearer advantages.
(3) The difference of this work from the following work in the case where the causal graph is known or unknown could be more clearly articulated, ‘[23] Kailash Budhathoki, Lenon Minorics, Patrick Bloebaum, and Dominik Janzing. Causal structure based root cause analysis of outliers’

---

> ### Author Rebuttal · Authors · 2025-07-31
>
> > Addressing weaknesses
>
> 1. You are right, the polytree assumption is indeed restrictive (though note that it does not restrict nodes to having only a single parent, it rather means there are no undirected cycles, but you are right that for Theorem 3.9 we additionally assume only a single parent). In subsection 3.5 and Appendix D we give some justification for the application of our algorithms even in cases where this assumption is violated, and in all experiments we do not restrict the causal graph to be a polytree, and both of our algorithms show competitive performance. Note also that the only other root cause analysis approach which operates in the setting we study (Li et al, 2025, “Root cause discovery via permutations and Cholesky decomposition”), the method we refer to as “Cholesky” in the Experiments section, assumes that the causal model is linear, while our approach does not. As such, we still believe that our work is an important contribution to the current state of root cause analysis research. To better highlight are discussion of the polytree assumption, we will move some of the discussion from Appendix D into the main text as there we show that we expect SCORE ORDERING to fail only rarely when the polytree assumption is violated (at least in the linear setting).
>
> 2. This is a good observation. Generating challenging synthetic data that is both realistic, but whose simulation parameters are interpretable (so that we can study why and when our algorithms fail), is often a challenge, but note that we do see considerable performance differences between even the top performing algorithms when applied to real-world (and semi-synthetic) datasets in Appendix E. Nonetheless, we will expand the synthetic data experiments to address this shortcoming. In particular, we propose comparing the performance of each of the algorithms which depend on being given the causal graph (i.e. Traversal, Counterfactual, SMOOTH TRAVERSAL, and Circa) after introducing an increasing number of random edge additions/deletions/reversals to the graph they are given, simulating the case where the assumed graph does not perfectly match the true graph. This will not affect the performances of SCORE ORDERING and Cholesky which do not rely on the causal graph, but should allow us to test how robust SMOOTH TRAVERSAL is to misspecification of the graph compared to the other algorithms, and to assess at what point it is preferable to use a method which does not rely on the causal graph, such as SCORE ORDERING, compared to one that does.
>
> 3. Thank you for raising this point as it is of particular importance to our work. Budhathoki et al not only assumes that you know the causal graph, but assumes you know the full structural causal model. Such an assumption is extremely prohibitive as in practice the structural causal model cannot be estimated from either observational or interventional data. We will better emphasise the differences of our work from that of Budhathoki et al in the introduction. Note that we devote Appendix A to showing how our work relates to that of Budhathoki et al even though we do not need the structural causal model (or even the causal graph in the case of SCORE ORDERING).
>
> > Addressing questions
>
> 1. Please see answer to point 2 in "Addressing weaknesses" above
> 2. Please see answer to point 3 in "Addressing weaknesses" above
> 3. The estimator in Equation 5 corresponds to applying the $\tau$ feature map to each sample in the normal period, and to the single sample from the anomalous distribution, and counting the number of points which are equal or greater in the normal period, to the anomalous sample, followed by taking the negative logarithm. In other words, it is the negative log of the quantile with respect to the empirical distribution. We will include this additional description following Equation 5 in the text to aid with clarity.
> 4. This is a good question. Although we have focused on the case where there is only a single anomalous sample, as this is a more difficult setting, the theory we present is equally suitable in the setting where one has more than one anomalous sample. One must instead define multi-dimensional feature maps ($\tau$ in Equation 3), $\tau: \mathcal{X}^{m} \rightarrow \mathbb{R}$ where $m$ is the number of anomalous samples. In this case, in effect one treats the collection of anomalous samples as being a single multi-dimensional sample and so the subsequent theory is unchanged. The advantage in this setting is that an appropriately defined feature map can enable the detection of more general distribution changes than are possible from a single sample. For example, if the anomalous distribution for a variable were to reduce in variance, but be unchanged in mean, each individual sample drawn from it would not appear anomalous, but together multiple samples would have an “unusually” low variance, and so collectively the samples would receive a high anomaly score for an appropriately defined feature map. Alternatively, if one assumes that samples are drawn iid from the anomalous distribution, as is often assumed in RCA literature, then one can define joint anomaly scores (joint over the $m$ samples) for each variable, similarly to Equation 7 in subsection 3.2 (or in the arXiv version of Budhathoki et al, which you have cited above) by taking the convolution of the individual sample anomaly scores. Although defining a multi-dimensional feature map would be preferable, another option as a simple heuristic in practice could be to apply our algorithms to each sample individually, and then pick a variable as root cause according to a rule such as majority-vote.

---

> > ### Comment · Reviewer_RvMn · 2025-08-05
> >
> > Thank you to the authors for the detailed review. I agree with point 2 in the “Addressing Weaknesses” section, adding some faults to the causal graph could help demonstrate the limitations of methods that require a graph. I also appreciate point 4 in “Addressing Questions,” which raises interesting propositions for cases involving collections of anomaly points.
> >
> > I hope the authors can further clarify in the experimental section which methods require a graph and which do not.
> >
> > Thank you again for your valuable contribution to this work.

---

### Official Review · Reviewer_Mb8b · 2025-06-20

**Clarity:** 3
**Significance:** 2
**Originality:** 3
**Rating:** 4
**Confidence:** 4

**Summary:**

This paper proposes two simple and efficient algorithms, Smooth Traversal and Score Ordering, to perform root cause analysis (RCA) of anomalies when the causal structure is either known (a tree) or unknown. It builds on the idea of modeling anomalies as soft interventions in a CBN and focuses on the regime where only a single sample from the anomalous setting is available. The experiments on synthetic data and PetStop dataset show the value of the proposed methods.

**Questions:**

1. Could you include a comparison with methods such as $\epsilon$-Diagnosis and RCD in a regime where more than one anomalous sample is available (e.g.,100 samples)?

2. For the PetShop dataset (Tables 1 and 3), how many nodes are typically flagged as potential root causes? What proportion of the total graph do these represent?

I find the paper really intriguing and I think it has value. I am willing to increase my score if the authors can address my questions.

**Ethical Concerns:**

["NO or VERY MINOR ethics concerns only"]

**Final Justification:**

I have considered the discussion with the authors and have decided to increase my score Borderline Accept. I would have wanted transparent discussion on when the anomaly score is tied and evaluation when more than one anomalous sample is given.

**Limitations:**

Yes

**Quality:**

2

**Strengths And Weaknesses:**

Strengths:

- The theoretical contributions are well-developed and remove the need for selecting arbitrary p-value thresholds.
- The proposed use of marginal anomaly scores in both known and unknown causal settings is elegant and supported by strong theoretical guarantees.

Weaknesses:

- Limited evaluation on diverse real-world scenarios
- I could not find any link to the source code.

Detailed Feedback:

I found the theoretical contributions really interesting, especially the principled treatment of anomaly scores and the avoidance of arbitrary thresholds. This is a refreshing alternative to many existing RCA approaches. However, I have concerns about the limited scope of the empirical evaluation and the real-world applicability of the proposed methods.

The justification for excluding comparisons with RCA methods like $\epsilon$-Diagnosis and RCD is that those methods assume access to multiple anomalous samples, whereas this work operates in the single-sample regime. While this is a valid distinction, in many real-world systems it is often feasible to collect a small number of anomalous samples (e.g., 50–100). Therefore, I believe the comparison with the existing approaches in the area of multiple anomalous samples is crucial for the work.

The motivation for this suggestion stems from a concern that, in complex systems, even with access to more than a single anomalous sample, the anomaly scores for most (if not all) nodes may remain very low. In such cases, the algorithm might end up flagging a large number of nodes as anomalous, hence diminishing its practical utility. This issue is evident in Table 3 of the Appendix, where the proposed method achieves only 14% top-1 recall. Furthermore, the justification provided in Tables 1 and 2—treating multiple nodes with identical scores as equally valid root causes—seems unrealistic in real-world scenarios. This raises an important question: how many nodes (and what proportion of the total) are typically flagged as potential root causes? As it stands, the evaluation does not fully convince me that the proposed methods can be reliably applied in large, noisy, or complex environments where more than one anomalous event or a broader root cause structure is involved.

---

> ### Author Rebuttal · Authors · 2025-07-31
>
> > Addressing weaknesses
>
> 1. You are right that we could do more to evaluate on a wider variety of real-world scenarios. To expand our evaluation to more real-world scenarios, we will in addition to the experiments we have already included using the PetShop dataset and the ProRCA package, we will evaluate each of the algorithms on the Sock-shop 2 dataset (Pham et al, 2024, “Root Cause Analysis for Microservice System based on Causal Inference: How Far Are We?”), and additional Top-1 and Top-3 recall performance tables in the Appendix.
>
> 2. We apologise for its omission upon submission. We would have shared it as part of our rebuttal if it were not this year’s changes to the rebuttal format. We will include a link to the source code should the paper be accepted.
>
> > Addressing detailed feedback
>
> You raise an important point regarding our algorithms flagging up many potential root causes. This occurs due anomalies receiving tied anomaly score estimates, however, the issue actually stems not from too few samples in the anomalous period (but too few samples in the normal period), nor from anomaly scores being too low (in fact rather when anomalies are too strong). As noted on line 105 below Equation 5, the maximum (estimated) IT anomaly score that can be achieved in a sample of size k is $\log k$. This corresponds to the case where the sample being scored is more anomalous than any sample observed in the normal period. As such, a very strong anomaly at the root cause may not only receive the maximum anomaly score, but also cause anomalies in its descendants which also receive the maximum score (even if the caused events are considerably less rare) if the sample size in the normal period is too low. For example, if one has 1000 samples in the normal period, a 1/1000 anomaly event will receive the same (maximum) estimated score as an 1/10,000 event. Note that this is not an issue with the anomaly score itself, but with the estimates given by the estimator in Equation 5. For example, if one had a parametric estimate of the tail of the distribution, then tied score estimates would not occur. Tied score estimates are less of an issue in settings where many samples are available from the normal period, but are prohibitive in our real-world PetShop experiments wherein there are no more than ~600 normal period samples (and often far fewer due to missingness). Regarding low anomaly scores, note that we assume that we are performing root cause analysis due to a (sufficiently) large anomaly being detected at a target node. Such an anomaly could not be explained if all nodes received a low score. Finally, it is important to note that in noisy systems the problem is actually also less pronounced. In a highly deterministic system, anomalies will propagate (close to) perfectly from root cause to descendants, resulting in anomaly scores which decay slowly along paths. In more noisy systems, anomaly scores decay rapidly along paths, further reducing the risk of descendants receiving the maximum possible estimated score even if the root cause does. Interestingly then our results may be expected to actually perform better in larger, more noisy systems when more normal period data is available.
>
>
> > Addressing questions
>
> 1. This is an important point you are right to raise as although the single-sample setting is more challenging, and is the focus of the work, it is nonetheless true that in many settings one may have access to more than one sample. Note that we do compare our algorithms to RCD and epsilon-Diagnosis in Tables 1 and 2 on the PetShop data wherein all methods that require multiple anomalous samples are given access to between two and five samples in the anomalous period (depending on how many samples were missing in the dataset). However, we will extend the comparisons made using synthetic data to RCD and epsilon-diagnosis as you have suggested. It is important to note, however, that RCD is a conditional independence-based and as such performs poorly when the sample size in the anomalous period is too small. For example, Figure 2 in the supplementary material for the paper presenting RCD (Ikram et al, 2022, ​​Root Cause Analysis of Failures in Microservices through Causal Discovery) shows that in synthetic data the recall for RCD only becomes competitive (>0.7) once there are thousands of samples for the anomalous period, whereas both SCORE ORDERING and SMOOTH TRAVERSAL exceed this recall in our own experiments (Figure 1 of our paper) for only a single anomalous sample, but we agree a combined comparison with our own algorithms should certainly be done nonetheless.
>
> 2. This is certainly an important question, and as noted above, the issue is exacerbated in the PetShop dataset due to the relatively low sample size in the normal period (due to high missingness) and relatively strong anomalies at the root cause. Below is a table summarising the average number of nodes which receive the same maximum estimated IT anomaly score, therefore will all be flagged by SCORE ORDERING, split according to issue type in the PetShop dataset. Note that the PetShop graph contains 41 nodes. We will include this table in Appendix E alongside Table 3 and 4. The table reveals what a challenge tied maximum estimated scores are in the PetShop dataset.
>
> | Scenario          | Metric       | Avg Nodes | Avg Proportion |
> |--------------------|-------------|---------------|----------------|
> | high_traffic       | Latency      | 18.26         | 41.50%         |
> | high_traffic       | Availability | 16.46         | 37.41%         |
> | low_traffic        | Latency      | 9.96          | 23.72%         |
> | low_traffic        | Availability | 6.54          | 15.57%         |
> | temporal_traffic1  | Latency      | 5.75          | 14.02%         |
> | temporal_traffic1  | Availability | 7.00          | 17.07%         |
> | temporal_traffic2  | Latency      | 5.75          | 14.02%         |
> | temporal_traffic2  | Availability | 7.12          | 17.38%         |

---

> > ### Comment · Reviewer_Mb8b · 2025-08-04
> >
> > Thank you to the authors for providing such a detailed response.
> >
> > You mentioned that the number of normal samples leads to multiple nodes receiving tied anomaly scores. Could you clarify why that happens? Specifically, what is the relationship between the number of normal samples and the accuracy or resolution of the scoring function?
> >
> > Secondly, you mentioned that failure propagation does not lead to increasing anomaly scores among descendants. Why cannot there be a model where failure effects propagate to descendants with increasing intensity? In that case, the anomaly score would accumulate as it moves downstream, and the last node could be flagged as the root cause. That said, what if the system contains a mixture of nodes—some that amplify failure signals, and others that decays them? In most real-world systems, behavior is neither purely deterministic nor purely random; rather, certain nodes may exacerbate a failure while others suppress it. How do you expect the SMOOTH TRAVERSAL to perform?
> >
> > Also, thank you for including the table showing the proportion of nodes flagged by the algorithm—it aligns well with my expectations. You mentioned the average number of nodes—may I ask what the average is taken over (number of runs perhaps)? Lastly, do you expect these number of nodes with tied scores to change if you included more anomalous or normal samples? Would it be possible to test that?

---

> > > ### Author Response · Authors · 2025-08-05
> > >
> > > > You mentioned that the number of normal samples leads to multiple nodes receiving tied anomaly scores. Could you clarify why that happens? Specifically, what is the relationship between the number of normal samples and the accuracy or resolution of the scoring function?
> > >
> > > This is a good question and is important to clarify. Multiple nodes receiving the same estimated anomaly scores is due to the simple estimator used in Equation 5. For this estimator, if the sample size in the normal period is $k$ then the estimated score on a sample in the anomalous period takes values in {$\log i : i \in$ {$1, \dots, k + 1$}}, i.e. for sample size $k$, the estimated score can take $k + 1$ possible values. Two nodes receive the same estimated score if their corresponding values (after applying their respective feature maps) happen to fall in exactly the same quantiles as each other in their empirical respective distributions. If the anomaly score for a given variable $X$ is surjective (i.e. $S_{X}: \mathcal{X}\to [0, \infty)$, is surjective) then asymptotically the probability that two anomalies receive the same score is zero as the discretisation of the score estimate disappears as $k$ grows. Note that the tying of scores does not depend at all on the sample size in the anomaly period. The case where low sample size in the normal period is most notable is when the anomaly at the root cause is very strong. For example in the PetShop dataset, as noted, the normal period sample size never exceeds ~600 (and is often much smaller), which means the maximum value that an estimated score can obtain is < $\log 600 \approx 6.4$. Suppose the true (not estimated) anomaly score for the anomaly at the root cause was 8.0, and this caused an anomaly in one of its children with true score of 7.0. The estimated scores for both the root cause and the child would both be 6.4 as their true scores exceed the maximum score which can be estimated with Equation 5. In this sense, ties occur because the estimated score is “saturated” and only by increasing the sample size in the normal period would the two scores become distinguishable. There would be no such problem if we used, for example, a parametric estimator for the tails of the distributions in place of Equation 5.
> > >
> > > With this in mind, we do think it is important to re-emphasise, however, that while small sample sizes in the normal period can pose an issue due to tied score estimations when using Equation 5, the sample requirements of our algorithms are very modest compared to current state-of-the-art approaches. This is not only because our methods are the only ones which can operate with a single anomalous sample in the non-linear setting, whereas methods such as RCD (Ikram et al, 2022, cited above) require >1000 samples, but because even in the normal period, existing methods either depend on large sample sizes or parametric assumptions. Large sample sizes in both the normal and anomalous period are required by the conditional independence-based RCA approaches, such as RCD, CauseInfer (Chen et al, 2014), AutoMap (Ma et al, 2020) and MicroCause (Meng et al, 2020), as well in NN-based approaches such as CausalRCA (Xin et al, 2023). Methods which have lower sample size requirements typically rely on parametric assumptions such as Circa (Li et al, 2022, cited in the paper fully), the method we refer to as “Cholesky” (Li et al, 2025, cited in the paper) and MicroDiag (Wu et al, 2021) which all assume linearity. Any similar parametric assumption for the estimation of the tail of the distribution would also solve the tied score estimate issue in our case too, and for larger sample sizes, we need no such assumptions, so those methods have more demanding sample requirements than our own.

---

> > > ### Author Response · Authors · 2025-08-05
> > >
> > > > may I ask what the average is taken over (number of runs perhaps)?
> > >
> > > Yes, sorry, we were not entirely clear in our reply. The average is taken over the replicates of each injected fault for each “type”, e.g. over the replicates for “latency faults” for low user-traffic to the application. This directly matches the replicates over which the average recalls (as displayed in Tables 1-4 in Appendix E) are taken. We did this so that the recall values displayed there are directly comparable to the averages displayed in the new table provided in our reply above. For the particular details of what each of these fault “types” mean, we would refer you to the original paper presenting the dataset (Hardt et al. “The PetShop Dataset -- Finding Causes of Performance Issues across Microservices”).
> > >
> > > > Lastly, do you expect these number of nodes with tied scores to change if you included more anomalous or normal samples?
> > >
> > > Thank you for giving the chance to clarify these points, they are important. Please see our reply to your first point above for the full details, but in brief summary: the number of anomalous samples does not influence whether nodes have tied estimated scores, but the number of normal samples does. Having more normal samples would reduce the probability of two nodes receiving the same score (and asymptotically the probability is zero) because the discretisation of the estimate given by Equation 5 would disappear, and two nodes only receive the same score if their values happen to fall within exactly the same discrete quantile. We hope this helps clear up why and when estimated score ties occur, and why PetShop in particular suffers from this problem (due to small sample size in the normal period, due to missingness, and due to strong anomalies at the root causes)!

---

> > > > ### Comment · Reviewer_Mb8b · 2025-08-05
> > > >
> > > > I would like to thank the authors for an engaging and thoughtful discussion. I have learned a lot from their insights and appreciate the depth of their experience. I believe the paper presents valuable ideas that should be shared the broader community. Accordingly, I am increasing my score to Borderline Accept.
> > > >
> > > > That said, I strongly encourage the authors to incorporate some of the key points raised during the discussion with me and the other reviewers into the final version of the paper. In particular, I found the argument regarding the availability of a large number of normal samples especially compelling. In many real-world scenarios, collecting normal data is far more feasible than collecting anomalous examples. Recent work has increasingly built on this observation, aiming to reduce reliance on anomalous data and instead use normal samples for effective RCA. An approach that improves with more normal samples can be especially impactful in such settings.
> > > >
> > > > Finally, I urge the authors to clarify the experimental setup and, if the paper is accepted, to release the code and datasets as mentioned earlier. Doing so would significantly enhance the reproducibility and impact of the work. I hope these suggestions will be addressed in the camera-ready version.
> > > >
> > > > Thank you again for your contribution.

---

### Official Review · Reviewer_P4jd · 2025-06-22

**Clarity:** 3
**Significance:** 3
**Originality:** 3
**Rating:** 4
**Confidence:** 3

**Summary:**

This paper presents a novel approach to Root Cause Analysis (RCA) for outliers, focusing on the challenging yet practical scenario where only a single anomalous data sample is available and the causal graph may be unknown. The authors introduce two efficient methods based on information-theoretic anomaly scores. For cases with a known tree-structured causal graph, the SMOOTH TRAVERSAL algorithm identifies the root cause by finding the largest increase in anomaly score between a node and its parents. When the graph is unknown, the SCORE ORDERING heuristic is proposed, providing a causally-justified method for shortlisting root causes by simply ranking variables by their marginal anomaly scores. Experiments demonstrate that SMOOTH TRAVERSAL is competitive with more complex, state-of-the-art methods, while SCORE ORDERING provides a robust and simple heuristic.

**Questions:**

Why does Assumption Continuity in section 3.2 (line 182) hold? More justification will help.

**Ethical Concerns:**

["NO or VERY MINOR ethics concerns only"]

**Final Justification:**

I have read authors' rebuttal and will remain my initial evaluation

**Quality:**

3

**Strengths And Weaknesses:**

Strength
1. The methods proposed are simple but effective without strong assumption and requirement on the input. The proposed methods are practical and easy for implementation.
2. Through simplicity, the authors provide theoretical justification of the methods under certain assumptions. Moreover, the authors point out the key insights behind both methods as an anomaly is unlikely to cause a much larger one downstream.
3. The authors carry out empirical experiments on synthetic causal graphs with comparison to several state-of-art methods.

Weakness
1. The paper's primary theoretical guarantees for both the SMOOTH TRAVERSAL and SCORE ORDERING methods are developed under the assumption that the causal graph is a polytree (a tree-like structure). It would be great to see experiments on how the violation of the property impacts the performance of the proposed method.
2. The methodology is designed around the hypothesis that there is only a single root cause for the observed anomaly. It would be interesting to see how the results generalize if multiple anomalies exist.

---

> ### Author Rebuttal · Authors · 2025-07-31
>
> > Addressing weaknesses
>
> 1. Thank you for this suggestion, this is indeed an interesting question to address. It is worth noting that in all experiments presented in the manuscript (both synthetic and real-world), we do not constrain the causal graph to be a polytree and already both of our proposed algorithms have competitive performance. In particular, it will be difficult to detect any improvements in the performance of SMOOTH TRAVERSAL were we to constrain the causal graphs to be polytrees, however, it remains an open question as to how much SCORE ORDERING could benefit (and whether any benefit exceed that for the other algorithms against which we compare). We will therefore repeat the experiments in figure 1 for polytrees and collider-free polytrees and include two extra figures in the manuscript should it be accepted.
>
> 2. We agree this is definitely an interesting question! Please see the detailed reply we gave to reviewer D5Fu to their question three at the end of our reply to them as it concerns exactly the same question of how to generalise to the cases of multiple root causes (and apologies for sending you to look at our replies to other reviewers!)
>
> > Addressing questions
>
> 1. Thank you for raising this issue, we admit that it is often not clear which assumptions are required for which results. In the revised manuscript we will state next to each assumption exactly which Lemmas and Theorems rely on them. For the question of continuity in particular: we use continuity as a concise condition as it is sufficient to ensure conditional anomaly scores are statistically independent (Lemma 3.4). We need this for Theorem 3.6 later in the section and again for Theorem 3.8. However, continuity is not actually strictly necessary, more generally, any condition which would sufficient to ensure the conditional anomaly scores are statistical independent would do. We will clarify this in the text following the statement of the assumption.

---

### Official Review · Reviewer_D5Fu · 2025-07-14

**Clarity:** 2
**Significance:** 2
**Originality:** 2
**Rating:** 4
**Confidence:** 3

**Summary:**

The paper presents a method for Root Cause Analysis that is based on analyzing the anomaly scores of variables. Assuming a single abnormal variable and a causal graph that is a polytree, the paper develops algorithms for discovering the abnormal variable for cases when the causal graph is known and unknown. It also establishes a connection between the abnormality of conditional probabilities (soft interventions) and marginal/conditional anomaly scores based on statistical tests.

**Questions:**

1. What prevents the estimation of conditional anomaly scores as shown in Section 3.2?
2. Theorem 3.9: It is unclear whether the equations $S(x_j)-S(pa_j) \geq S(x_n)/2$, $S(x_i) - S(pa_i) \leq S(x_n)/4$ are derived from the structural property of the DAG being a tree, or if they are stated as assumptions.
3. How difficult would it be to generalize the result to multiple root causes? A brief discussion on this can be quite helpful.

**Ethical Concerns:**

["NO or VERY MINOR ethics concerns only"]

**Final Justification:**

My concerns have been addressed.

**Limitations:**

yes

**Quality:**

2

**Strengths And Weaknesses:**

Strengths:
1. The idea of identifying root causes based on anomaly scores is interesting. While the approach builds on [Budhathoki et al., 2022], the contributions are novel.
2. The paper offers rigorous theoretical results, with complete proofs in the appendix, and presents reproducible pseudocode. It also includes simulation results on randomly generated SCMs.
3. Overall, the paper is well-structured, and the writing is mostly clear.

Weaknesses:
1. I could not find any soundness/completeness results on the Smooth Traversal and Score Ordering algorithms when the causal graph is a polytree (lower-bound is only provided for tree structures), which is central to the paper’s contributions. The provided lower bound applies only to tree structures. Are there scenarios in which these algorithms are guaranteed to be sound and complete? Please let me know if I missed something.
2. Lines 56-58: I would not consider the polytree assumption to be "weak." Moreover, the methods also rely on assumptions about the anomaly scoring function, which should be clearly stated in the introduction.
3. Lines 134-135: What's the definition of N and id?
4. In Section 3.3, is the Continuity assumption still required for marginal anomaly scores? This should be clarified at the beginning of the subsection.
5. Line 248: Instead of referring to "this final assumption," please name the specific assumption to improve clarity.
6. I recommend including concrete examples to illustrate key graphical concepts (e.g., polytree vs. tree), as well as the notions of marginal/conditional anomaly scores and how the two main algorithms perform root cause selection.
7. The conclusion section appears incomplete.

---

> ### Author Rebuttal · Authors · 2025-07-31
>
> > Addressing weaknesses:
>
> 1. Thank you for raising this point. Let’s say a root cause analysis algorithm is sound if every variable it returns is a root cause, and say it is complete if every root cause among the input variables is returned. As our algorithms take marginal anomaly scores, which are only defined with respect to particular samples, it is not possible to define soundness/completeness independently of sample variation during the anomalous period. In this sense we cannot guarantee soundness/completeness. Literature on algorithms for causal discovery often define an oracle with respect to which soundness/completeness are defined in order to circumvent this caveat, but it is not clear what a suitable oracle would be in our setting. Nonetheless, Theorem 3.10 can be interpreted as a soundness result for SCORE ORDERING as it shows that if the largest anomaly score exceeds the next largest by a large amount, then the probability that the top-scoring variable is not the root cause is small—i.e. SCORE ORDERING is sound with high probability in this case. For SCORE ORDERING to be complete we would need to show that any root cause variable will have (with high probability) a larger score than non-root causes (as otherwise SCORE ORDERING will not return every root cause with high probability). As such, a priori we cannot expect completeness without further assumptions as in deterministic causal systems, anomalies are perfectly propagated, so downstream nodes will receive the same score as the root cause. To guarantee the anomaly score decays, we need to postulate constraints on the minimal noise level. For instance, for a linear Gaussian cause-effect relation, $X \to Y$, with Pearson correlation $\rho$, the $z^2$ score of $Y$ is with high probability close to $\rho^2$ times the $z^2$ of $X$, which can be translated into a decay of IT anomaly scores by recalibration. We are happy to include such model-dependent guarantees for score differences, but we currently have no idea for a concise general constraint on the noise level. For SMOOTH TRAVERSAL, we have shown (Lemma 3.7) that the score of a non-root cause is unlikely to exceed that of its parent by a large amount, i.e. if the score difference is large, SMOOTH TRAVERSAL is likely to be sound. Completeness would mean that every root cause exceeds the score of its parents by a large amount, which also requires additional assumptions. It is not obvious to find an assumption that does not trivially imply what is supposed to be shown.
>
> 2. Our apologies, you are quite right of course that the polytree assumption is not weak. Indeed on lines 282–283 we note that it is a strong assumption. Lines 56-58 were rather poorly phrased: the intention was to say that the assumptions on the anomaly scores are weak, first applied to cause-effect pairs and then to polytrees, but we accept that this was unintentionally misleading. We will remove the word “weak” and state explicitly that our results require so-called “information theoretic (IT)” anomaly scores as well as additional assumptions on them.
>
> 3. N is an independent noise variable, as in a structural causal model, and id is the identity function. We will alter the text to define both as suggested.
>
> 4. Thank you for raising this issue, we admit that it is often not clear which assumptions are required for which results. In the revised manuscript we will state next to each assumption exactly which Lemmas and Theorems rely on them. For the question of continuity in particular: yes we still use the continuity assumption for the results concerning marginal anomaly scores in subsection 3.3. This is because to obtain the bound on the p-value in Theorem 3.8, we need (in addition to Lemma 3.7) Theorem 3.6 which itself follows from Lemmas 3.4 and 3.5 under the assumption of continuity in the previous subsection. However, continuity is not strictly necessary: we use it as a concise condition as it is sufficient to ensure conditional anomaly scores are statistically independent (Lemma 3.4). More generally, any condition which is also sufficient would do. Note we do not use continuity in subsection 3.4 for the results concerning using marginal anomaly scores for SCORE ORDERING.
>
> 5. On line 248 the assumption in question is that we know the causal graph, stated in the previous sentence, but we agree this should be clarified: we will change the text accordingly.
>
> 6. Thank you for the recommendation, we agree that this would improve understanding of the paper. To illustrate the difference between conditional and marginal score we will introduce an example where we consider a linear Gaussian cause-effect pair. This is because in the linear Gaussian setting, marginal IT anomaly scores can be derived via re-calibrating $z^2$ scores on the variables themselves, while conditional scores are derived from $z^2$ scores of the regression residual. To illustrate the key graphical concepts, we will add an additional figure showing an example of a polytree, collider-free polytree and a DAG, and highlight their differences.
>
> 7. Thank you for raising this, you are right, the conclusion is unhelpfully brief. This was an unfortunate mistake arising from cutting text to meet the page limit. We will expand the conclusion, and suggest the following: “We have explored several directions in which the practical limitations of RCA can be addressed: first by avoiding estimation of conditional distributions in the setting that the causal graph is known, and second by avoiding needing the causal graph at all, in the setting that it is a polytree. To do so, we leverage information theoretic (IT) anomaly scores, and prove general laws about their typical decay along causal paths. Using our results, we then propose two novel algorithms: SMOOTH TRAVERSAL and SCORE ORDERING, give probabilistic guarantees for the correctness of their outputs, and show they have competitive performance in both synthetic and real-world data despite their simplicity. To our knowledge, our work is the only which provides guarantees for RCA in the non-linear setting from a single anomalous point without requiring knowing the full structural causal model.”
>
> > Addressing questions:
>
> 1. Estimating conditional anomaly scores is hard as it requires estimation of conditional probabilities, which is statistically ill-posed, as noted on lines 111-112 at the top of Section 3. We will clarify this point at the end of subsection 3.2 to motivate the approach followed in subsection 3.3.
>
> 2. We agree this is unclear. The inequalities are neither derived from structural properties nor are they assumptions, rather, when the inequalities hold, the root cause is the unique node which maximises the score difference with its parent from among the ancestors of the target node. The theorem then establishes a lower bound on the probability that the inequalities hold (and thus the root cause is correctly identified) subject to the structural assumptions on the DAG and the previously stated assumptions on the anomaly scores. We will rephrase the theorem to avoid confusion and suggest the following: “Let the causal DAG for $X_1,\dots,X_n$ be a collider-free polytree and node $j\in \{1,\dots,n\}$ be the unique root cause of the anomaly $x_n$, then the following inequalities hold with probability at least $1 - (n-1)\cdot e^{-\frac{1}{4}S(x_n)}$: $S(x_j) - S(pa_j) \geq S(x_n)/2$, $S(x_{i}) - S(pa_i) \leq S(x_n)/4 \forall i\neq j$, and hence root cause $j$ is the unique node that maximises the anomaly score difference with its parent.”
>
> 3. This is a very good question and we agree it’s important that we include a discussion as suggested. The results in subsection 3.4, concerning SCORE ORDERING, already apply to the case of having multiple root causes (i.e. more than one variable whose causal mechanism is corrupted for the single sample from the anomalous regime). This is because Theorem 3.10 is stated in terms of rejecting the hypothesis that none of the top-k highest scoring anomalies, as such, rejecting this hypothesis does not imply that there is only a single root cause among the top-k, but rather at least one. SCORE ORDERING therefore remains a suitable approach in this more general setting. We can say something similar concerning SMOOTH TRAVERSAL by slightly altering the null hypothesis given at the beginning of subsection 3.2 (lines 174-175). Instead of rejecting the hypothesis that all causal mechanisms worked as normal, except possibly the one for $X_j$, we can reject the hypothesis that all causal mechanisms worked as normal except for possibly $X_j$ and $X_l$. Following exactly the same steps as before in subsections 3.2 and 3.3 (i.e. defining the joint anomaly score but now excluding both $X_j$ and $X_l$ from $S_{\rm sum}$), we will arrive at a corresponding version of Theorem 3.8 wherein the bound on the likelihood is weakest if we infer the top two variables with the largest score difference from their parents as root causes. Similarly we could do the same for three root causes and so on. In other words, Theorem 3.8 justifies taking the top-k variables with the largest score differences from their parents whenever one suspects there are k root causes. The important caveat here is that while this approach for selecting k is suitable for small k, one incurs a larger multiple testing burden as k increases, and this would need to be accounted for. It would also be possible to adapt Theorem 3.9, but this is more intricate. In a similar spirit to SCORE ORDERING, when k is unknown, one could choose the minimum k for which the likelihood bound in Theorem 3.8 is now “acceptably” weak for some combination of k variables.

---

> ### Comment · Reviewer_D5Fu · 2025-08-07
>
> Thanks for the explanations. My concerns have been addressed. For weakness 1, I was initially expecting a result similar to Theorem 3.9, but for polytrees. Based on the authors' response, I now see that such a result may be difficult to establish without stronger assumptions about the causal model. That said, I agree it would be helpful to include some concrete examples illustrating when the proposed algorithms are guaranteed to be correct (such as the 2-node example mentioned by the authors) and when they can fail.
>
> Based on the response, I have increased my score.

---

### Author Response · Authors · 2025-08-08

We agree with the reviewers that the polytree assumption is strong, and that the principle of non-increasing IT anomaly scores along causal paths does not always hold outside this setting. However, fair comparisons to existing methods should take into account their own limitations, such as reliance on conditional independence testing, which often requires either large sample sizes, strong parametric assumptions, or both. In our applications, these limitations have been shown to be severe. Our methods perform competitively in empirical evaluations even when the polytree assumption is violated, and to the best of our knowledge, our results are the only non-parametric guarantees for root cause analysis that apply in the single anomalous sample setting. While the principle of non-increasing IT anomaly scores (Lemmas 3.2, 3.7, B.1, and B.2) does not solve all root cause analysis problems, we believe its generality and simplicity to be an important contribution to the field.

---

### Decision · Program_Chairs · 2025-09-17

**Decision:**

Accept (poster)

**Comment:**

This paper addresses the problem of root cause analysis (RCA) in the single-sample setting, introducing two efficient methods based on information-theoretic anomaly scores: Smooth Traversal (for known polytrees) and Score Ordering (for unknown graphs). The theoretical development is solid, with proofs and guarantees under specific assumptions, and the algorithms are simple yet competitive with more complex baselines.

Reviewers highlighted the novelty of the approach, the clarity of the theoretical contributions, and the practical relevance of studying RCA with limited data. Concerns focused mainly on the restrictiveness of the polytree assumption, the limited empirical evaluation on diverse real-world settings, the need for clearer exposition and broader comparisons, and insufficient literature review (some recent works in this direction were not discussed). The authors’ rebuttal addressed many of these points, clarifying assumptions, promising expanded experiments, and strengthening the discussion of limitations. Several reviewers increased their scores after the discussion.

Overall, while some limitations remain, the paper makes a timely and meaningful contribution to RCA research. I recommend Accept, as the theoretical insights and methodological simplicity outweigh the concerns, and the paper is likely to stimulate further work in this area.